# Graph Alignment via Dual-Pass Spectral Encoding and Latent Space Communication

**Maysam Behmanesh** [1]   **Erkan Turan** [1]   **Maks Ovsjanikov** [1]

## Abstract

Graph alignment, the problem of identifying corresponding nodes across multiple graphs, is fundamental to numerous applications. Most existing unsupervised methods embed node features into latent representations to enable cross-graph comparison without ground-truth correspondences. However, these methods suffer from two critical limitations: the degradation of node distinctiveness due to oversmoothing in GNN-based embeddings, and the misalignment of latent spaces across graphs caused by structural noise, feature heterogeneity, and training instability, ultimately leading to unreliable node correspondences. We propose a novel framework employing a dual-pass encoder to inject high-frequency discriminability into node features, paired with a geometry-aware functional map module that learns bijective and isometric transformations to align latent spaces while acting as a low-pass filter on correspondences, enforcing smoothness and robustness as a structural prior in map space. Extensive experiments on graph benchmarks demonstrate that our method consistently outperforms existing unsupervised alignment baselines, exhibiting superior robustness to structural inconsistencies and challenging alignment scenarios. The implementation is available at https://github.com/maysambehmanesh/GADL.

## 1. Introduction

Graph alignment, also referred to as network alignment or graph matching, is a fundamental problem in machine learning, concerned with identifying a correspondence between the nodes of two graphs such that structurally similar or semantically equivalent nodes are matched.

[1]LIX, Ecole Polytechnique, IP Paris. Correspondence to: Maysam Behmanesh < behmanesh@lix.polytechnique.fr>.

*Proceedings of the 43$^{rd}$ International Conference on Machine Learning*, Seoul, South Korea. PMLR 306, 2026. Copyright 2026 by the author(s).

Graph alignment arises in a wide range of application domains, including bioinformatics (e.g., protein interaction networks) (Liao et al., 2009; Singh et al., 2007), social network analysis (Li et al., 2018; Korula & Lattanzi, 2014), computer vision (Liu et al., 2022a; Chen et al., 2025; Wang et al., 2019), and natural language processing (Osman & Barukub, 2020; Guillaume, 2021). Due to its combinatorial nature, graph alignment is computationally challenging, often requiring approximation or heuristic algorithms.

Graph alignment methods are typically classified into three categories based on their alignment strategies: optimization-based, optimal transport–based, and embedding-based approaches. They also vary in the level of supervision required, ranging from unsupervised to semi-supervised, using partial node correspondences, and fully supervised methods. A detailed overview of these categories with related works is provided in Appendix A.

Embedding-based graph alignment methods encode graphs into low-dimensional node representations via Graph Neural Networks (GNNs) (He et al., 2025; Fey et al., 2020; Gao et al., 2021b), followed by alignment through transformations or joint learning with cross-graph regularization. Node matching is then performed using nearest-neighbor search or assignment algorithms, achieving better scalability than optimization-based alternatives.

These methods are typically formulated as unsupervised learning tasks, where ground-truth node correspondences across graphs are unavailable. Despite their computational advantages, these approaches face several inherent challenges that limit their effectiveness and reliability:

**1) Degradation of node distinctiveness in GNN embeddings.** While GNNs capture structural information by aggregating neighborhood features, this process inherently reduces node distinctiveness. This limitation is particularly problematic for graph alignment, where accurate correspondence identification depends on highly discriminative representations. As embeddings lose uniqueness, alignment becomes increasingly ambiguous and error-prone.

From a spectral perspective, standard GCN acts as an imperfect low-pass filter, suppressing discriminative high-frequency components while introducing noise for higher

frequencies. We propose a dual-pass encoder with two provably monotone spectral branches, strictly low-pass for structural context and strictly high-pass for node discriminability, and prove in Theorem 3.2 that the combined embedding is strictly more discriminative than either branch alone, while preserving neighborhood consistency (Section 3.2).

**2) Misaligned latent spaces across graphs.** In the absence of supervision, explicit constraints, or alignment-specific objectives, embedding-based methods struggle to produce comparable latent spaces across different graphs. Even with shared encoders, structural inconsistencies, feature heterogeneity, and training instability cause embeddings to occupy misaligned geometric spaces, mapping structurally identical nodes to distant latent regions (Figure 1).

This misalignment arises from multiple sources. First, structural inconsistencies, such as missing or noisy edges, distort neighborhood aggregation during message passing, leading to incompatible embeddings for otherwise corresponding nodes. Second, feature inconsistency across graphs, stemming from differences in user attributes or data domains, causes graph encoders to embed semantically equivalent nodes into disjoint subspaces. Lastly, GNN-based encoders often exhibit stochasticity in training; different random initializations can yield drastically different embeddings, even on fixed graph inputs (Moschella et al., 2023). Without explicit mechanisms to harmonize or align the latent spaces, these inconsistencies severely hinder cross-graph communication and undermine the reliability of node alignment.

Figure 1 illustrates the latent spaces learned by a 2-layer GCN on synthetic graphs. Panel (a) shows that identical graphs produce well-aligned embeddings, facilitating effective correspondence detection. However, panels (b) and (c) reveal that minor feature and structural inconsistencies cause corresponding node embeddings to diverge significantly, compromising alignment quality. Most critically, panel (d) shows that retraining the same model on identical graphs with different random initializations produces drastically different latent spaces, underscoring the instability and non-deterministic nature of learned representations.

In this paper, we introduce GADL, **G**raph **A**lignment with **D**ual-pass encoder and **L**atent space communication, which builds upon the Graph Autoencoder (GAE) framework (Kipf & Welling, 2016) to address the dual challenges of node distinctiveness degradation and latent space misalignment.

Our key insight is that rather than balancing these objectives purely in the feature space, we can enforce desirable properties through complementary filtering mechanisms in both feature and map spaces. GADL employs a dual-pass GCN encoder that combines low-pass and high-pass spectral filters to inject high-frequency discriminability into node features. The low-pass branch captures structural context

and the high-pass branch preserves node distinctiveness, yielding embeddings that are both structure-aware and discriminative. Crucially, this is paired with a geometry-aware functional map module that operates in the space of maps rather than features. This module learns bijective and isometric transformations to align latent spaces while acting as a low-pass filter on correspondences, enforcing smoothness and robustness as a structural prior in the map space.

Our key contributions can be summarized as:

1. We propose a unified framework that simultaneously addresses node distinctiveness degradation in feature space and latent space misalignment in map space.

2. We introduce a dual-pass encoder that combines low-pass and high-pass spectral filters to inject high-frequency discriminability into node features while preserving structural context.

3. We propose a geometry-aware functional map module that aligns latent spaces, while low-pass filtering correspondences, enforcing smoothness and robustness.

4. Extensive experiments on graph benchmarks show our method outperforms existing baselines in most settings with superior robustness to structural inconsistencies.

**Conflict of Interest Disclosure.** The authors declare no financial conflicts of interest.

## 2. Preliminaries

We formally define the problem of aligning attributed nodes from a source graph $\mathcal{G}_s$ to a target graph $\mathcal{G}_t$ in an unsupervised setting. The goal is to identify, for each node in the source graph, a corresponding node in the target graph.

**Definition 2.1** (Graph Alignment (GA)). Given two graphs $\mathcal{G}_s = (\mathcal{V}_s, \mathcal{E}_s, \mathbf{X}_s)$ and $\mathcal{G}_t = (\mathcal{V}_t, \mathcal{E}_t, \mathbf{X}_t)$, where $\mathcal{V}$ denotes the set of nodes, $\mathcal{E}$ the set of edges, and $\mathbf{X}_* \in \mathbb{R}^{N_* \times k_*}$ the associated node attributes (features), the *graph alignment problem* aims to find a one-to-one mapping $\pi : \mathcal{V}_s \to \mathcal{V}_t$ such that for each node $u \in \mathcal{V}_s$, $\pi(u) = v \in \mathcal{V}_t$ and $\pi^{-1}(v) = u$. The objective is to identify correspondences between nodes in $\mathcal{G}_s$ and $\mathcal{G}_t$ that preserve structural similarity and attribute consistency across the two graphs.

We assume the GA problem between two general graphs with different number of nodes ($|\mathcal{V}_s| \neq |\mathcal{V}_t|$) in an unsupervised setting, where no ground-truth node correspondences are available during training, and the alignment depends solely on the structural and attribute information of the graphs.

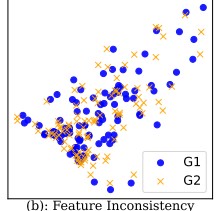 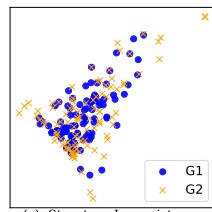 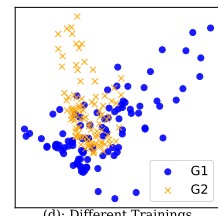

(a): Identical Graphs    (b): Feature Inconsistency    (c): Structure Inconsistency    (d): Different Trainings

*Figure 1.* Limitations of embedding-based graph alignment on synthetic data. $\mathcal{G}_1$ is a ring graph with 100 nodes and 2D random features. (a) $\mathcal{G}_2$ is identical to $\mathcal{G}_1$, yielding well-aligned embeddings. (b) Feature inconsistency introduced through Gaussian noise (std=0.2) causes divergence. (c) Structural inconsistency via 30% edge dropout distorts the embedding alignment. (d) Identical graphs with different training runs show embedding instability, highlighting unsupervised learning limitations.

## 2.1. Graph autoencoder for unsupervised node embedding

Graph autoencoders (GAEs) (Kipf & Welling, 2016) learn node embeddings in an unsupervised setting, generating low-dimensional representations that capture both node features and graph structure. Following the general principle of autoencoders, a GAE consists of two main components: an **encoder** $q_\theta(\mathbf{Z} \mid \mathcal{G})$ that maps the input graph $\mathcal{G} = (\mathcal{V}, \mathcal{E}, \mathbf{X})$, where $\mathbf{X} \in \mathbb{R}^{|V| \times k}$, into a latent embedding matrix $\mathbf{Z} \in \mathbb{R}^{|V| \times d}$, leveraging both graph structure and node features to learn meaningful representations; and a **decoder** $p_\phi(\mathcal{G} \mid \mathbf{Z})$ that reconstructs the original graph structure and node attributes from these latent embeddings, producing an approximation $\hat{\mathcal{G}}$ of the input graph. The model minimizes a loss combining a reconstruction loss $\mathcal{L}_{\mathrm{rec}}$ between $\mathcal{G}$ and $\hat{\mathcal{G}}$, and an optional latent space regularization term $\mathcal{L}_{\mathrm{reg}}$:

$$\mathcal{L} = \mathcal{L}_{\mathrm{rec}}(\mathcal{G}, \hat{\mathcal{G}}) + \lambda \mathcal{L}_{\mathrm{reg}}(\mathbf{Z}), \tag{1}$$

where $\lambda$ controls the regularization strength. This framework enables unsupervised learning of node embeddings that capture the intrinsic geometric structure of the graph, facilitating downstream alignment tasks.

In this framework, the encoder is typically implemented using a GNN $\phi(\mathbf{X}, \mathbf{S}; \theta) : \mathbb{R}^{N \times k} \to \mathbb{R}^{N \times d}$ with parameters $\theta$, which maps node features $\mathbf{X}$ and graph structure $\mathbf{S}$ (e.g., an adjacency or normalized Laplacian matrix) to latent node embeddings $\mathbf{Z}$. The decoder is typically a simple, non-parametric function that reconstructs the graph structure from the learned embeddings. A common choice is the inner product decoder, which estimates the adjacency matrix $\hat{\mathbf{A}}$ as $\hat{\mathbf{A}} = \mathbf{Z}\mathbf{Z}^\top$. This formulation assumes that the similarity between node embeddings reflects the likelihood of an edge, enabling the reconstruction of the graph topology directly from the embedding space.

## 2.2. Functional map on graphs

The functional map framework, originally proposed for 3D shape correspondence (Ovsjanikov et al., 2012), offers a compact and flexible approach that converts the problem of finding a complex node-to-node correspondence into learning a small, low-dimensional operator $C$ that aligns functions represented in a spectral basis. This paradigm naturally extends to graphs (Fumero et al., 2025; Behmanesh et al., 2024), where functions are defined on nodes, providing a powerful framework for comparing and aligning graph-structured data.

Building on the general framework of Deep Geometric Functional Maps (Donati et al., 2020), the functional map formulation is adapted to operate on graph-based latent representations through:

**1. Feature extraction.** Given a pair of graphs $\mathcal{G}_1$ and $\mathcal{G}_2$, each is associated with a set of descriptor functions, denoted by $\mathcal{F}_\theta(\mathcal{G}1)$ and $\mathcal{F}_\theta(\mathcal{G}_2)$, respectively. A descriptor function is a real-valued function defined on the nodes of a graph, either hand-crafted to capture structural information shared across graphs or learned via neural encoders, producing row feature matrices $\mathbf{F}_1$ and $\mathbf{F}_2$.

**2. Projection to spectral domain:** For each domain, the spectral basis $\Phi_*$ is computed via eigendecomposition of the normalized graph Laplacian $\mathbf{L} = \mathbf{I} - \mathbf{D}^{-\frac{1}{2}} \mathbf{A} \mathbf{D}^{-\frac{1}{2}}$. Descriptor functions are then projected onto the reduced spectral subspace $\Phi_* \in \mathbb{R}^{n_* \times r}$, spanned by the first $r$ eigenvectors, resulting in the spectral coefficients $\hat{\mathbf{F}}_1 = \Phi_1^\top \mathbf{F}_1$, and $\hat{\mathbf{F}}_2 = \Phi_2^\top \mathbf{F}_2$.

**3. Functional map estimation.** A functional map $\mathbf{C}_{12} \in \mathbb{R}^{r \times r}$ is then estimated by aligning the spectral descriptors between the two domains via the following regularized least squares objective:

$$\mathbf{C}_{12} = \arg \min_{\mathbf{C}} \|\mathbf{C}\hat{\mathbf{F}}_1 - \hat{\mathbf{F}}_2\|_F^2 + \alpha\|\Lambda_2 \mathbf{C} - \mathbf{C}\Lambda_1\|_F^2, \tag{2}$$

where the second term is the Laplacian commutativity regularizer, enforcing that $\mathbf{C}_{12}$ approximately commutes with the graph Laplacians to preserve spectral properties.

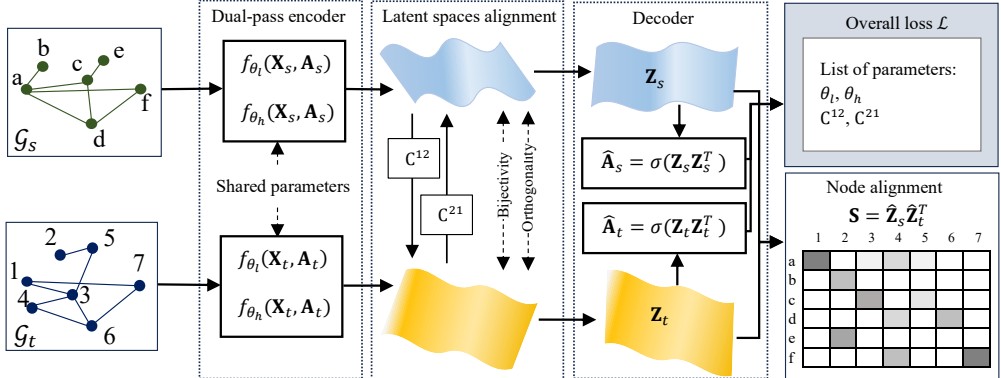

*Figure 2.* Overview of the proposed framework. Given input graphs, the model uses a dual-pass encoder with shared parameters to extract node embeddings. A regularized functional map module resolves latent space misalignment by enforcing structural constraints and enabling cross-space communication. A graph decoder reconstructs the inputs, and the model is optimized with an overall loss. Finally, alignments are estimated via cosine similarity and greedy matching.

## 3. Method overview

As established in the introduction, learning-based frameworks for graph alignment suffer from two fundamental limitations that significantly impair their performance: *loss of node distinctiveness* through feature aggregation and *misaligned latent spaces* in unsupervised cross-graph scenarios. In the following, we present the proposed framework that addresses these challenges through architectural innovations that preserve node distinguishability while enforcing embedding space alignment.

### 3.1. Overall framework

Given two graphs $\mathcal{G}_s = (\mathcal{V}_s, \mathcal{E}_s, \mathbf{X}_s)$ and $\mathcal{G}_t = (\mathcal{V}_t, \mathcal{E}_t, \mathbf{X}_t)$, the framework employs a dual-pass encoder with shared parameters $\theta$ to extract meaningful node representations. The encoder processes both graphs simultaneously, generating latent embeddings $\mathbf{Z}_s = f_\theta(\mathbf{X}_s, \mathbf{A}_s) \in \mathbb{R}^{|V_s| \times d}$ and $\mathbf{Z}_t = f_\theta(\mathbf{X}_t, \mathbf{A}_t) \in \mathbb{R}^{|V_t| \times d}$ by jointly encoding graph structure and node attributes. To address latent space misalignment, a regularized functional map module enforces structural constraints and enables communication between embedding spaces. Figure 2 provides a schematic overview.

### 3.2. Graph encoder

Given a graph $\mathcal{G} = (\mathcal{V}, \mathcal{E}, \mathbf{X})$, a graph encoder $\mathbf{Z} = f_\theta(\mathbf{X}, \mathbf{A}) \in \mathbb{R}^{|V| \times d}$ embeds each node $v_i \in \mathcal{V}$ into a latent vector $\mathbf{z}_i \in \mathbb{R}^d$, such that the embeddings of neighboring nodes are encouraged to be similar. While this property allows the encoder to capture the local graph structure effectively, it poses a significant limitation for graph alignment tasks by reducing the distinctiveness of individual nodes, an essential factor for accurately identifying corresponding nodes across graphs.

**Definition 3.1** (Ideal node embedding for graph alignment). An *ideal* node embedding for graph alignment achieves two properties: local consistency, where neighboring node embeddings are similar ($\max_{v \in V} \max_{u \in \mathcal{N}(v)} \|h_v^{(k)} - h_u^{(k)}\|$ is small), and global distinctiveness, where distinct nodes have sufficiently different embeddings ($\min_{v,w \in V} \|h_v^{(k)} - h_w^{(k)}\|$ is large). Graph alignment thus requires embedding nodes that balance a fundamental trade-off: preserving local similarities to capture structure while maintaining node distinctiveness for unique identification.

One of the simple yet effective graph encoders is the Graph Convolutional Network (GCN) (Kipf & Welling, 2017), which extends convolution to graph-structured data by aggregating neighboring node information to capture both features and structure. A single GCN layer is defined by $\mathbf{H}^{(l+1)} = \sigma\left(\tilde{\mathbf{D}}^{-\frac{1}{2}}\tilde{\mathbf{A}}\tilde{\mathbf{D}}^{-\frac{1}{2}}\mathbf{H}^{(l)}\mathbf{W}^{(l)}\right)$, where $\tilde{\mathbf{A}} = \mathbf{A} + \mathbf{I}$ is the adjacency matrix with self-loops, $\tilde{\mathbf{D}}$ is the degree matrix, $\mathbf{H}^{(l)}$ and $\mathbf{W}^{(l)}$ are the feature and weight matrices, and $\sigma(\cdot)$ is the activation function.

**Spectral interpretation of GCN:** In graph signal processing, the graph Laplacian is defined as $\mathbf{L} = \mathbf{D} - \mathbf{A}$, where $\mathbf{D}$ is the degree matrix and $\mathbf{A}$ is the adjacency matrix. The Laplacian can be decomposed as $\mathbf{L} = \mathbf{U}\mathbf{\Lambda}\mathbf{U}^\top$, where $\mathbf{U} = (\mathbf{u}_1, \ldots, \mathbf{u}_n)$ is the matrix of eigenvectors, and $\mathbf{\Lambda} = \mathrm{diag}(\lambda_1, \ldots, \lambda_n)$ is the diagonal matrix of eigenvalues. The normalized graph Laplacian is defined as $\mathbf{L}_{\mathrm{sym}} = \mathbf{D}^{-1/2}\mathbf{L}\mathbf{D}^{-1/2}$, whose eigenvalues $\lambda_i$ lie within the interval $[0, 2]$.

The GCN filter can be expressed as $\tilde{\mathbf{A}}_{\mathrm{GCN,sym}} = \mathbf{I} - \tilde{\mathbf{L}}_{\mathrm{sym}} = \mathbf{U}(\mathbf{I} - \tilde{\mathbf{\Lambda}})\mathbf{U}^\top$, with an associated frequency response function $p_{\mathrm{GCN}}(\tilde{\lambda}_i) = 1 - \tilde{\lambda}_i$. Since the eigenvalues satisfy $\tilde{\lambda}_i \in [0, 2]$, the response function $p_{\mathrm{GCN}}(\tilde{\lambda}_i)$ decreases as $\tilde{\lambda}_i$ increases, particularly over the range $[0, 1]$.

This behavior implies that the GCN filter primarily suppresses high-frequency components and thus acts as a low-pass filter in that region. However, for $\tilde{\lambda}_i > 1$, $p_{\text{GCN}}(\tilde{\lambda}_i)$ becomes negative, introducing noise and disrupting smoothness. This means GCN is not a completely low-pass filter and can degrade performance due to this issue.

**Node embedding via spectral filtering:** Low-pass filters preserve low-frequency components (small $\lambda_i$) and suppress high-frequency components, producing smooth embeddings where neighboring nodes have similar representations. Such embeddings are effective at capturing smooth, community-level structure within the graph. In contrast, high-pass filters preserve high-frequency components (large $\lambda_i$), emphasizing the differences between neighboring nodes. This leads to embeddings that capture distinctive, discriminative features, making the latent representations of nodes more distinct and farther apart from those of their neighbors.

**Dual-pass GCN encoder with spectral filtering:** In our proposed model, we design a dual-encoder architecture comprising two complementary GCN variants that exploit the spectral properties of graph signals. The architecture consists of: 1) a low-pass GCN encoder $\mathbf{Z}_l = f_{\theta_l}(\mathbf{X}, \mathbf{A})$ that produces smooth embeddings by aggregating information from neighboring node, and 2) a high-pass GCN encoder $\mathbf{Z}_h = f_{\theta_h}(\mathbf{X}, \mathbf{A})$, which highlights differences between a node and its neighbors, generating distinctive embeddings that effectively capture discriminative features. The final node representation is obtained through concatenation $\mathbf{Z} = [\mathbf{Z}_l \, \| \, \mathbf{Z}_h] \in \mathbb{R}^{|V| \times (d_l + d_h)}$.

Both encoders employ a unified spectral convolution framework with layer-wise propagation rule:

$$\mathbf{Z}_*^{(m+1)} = \sigma\left(\tilde{\mathbf{D}}^{-\frac{1}{2}} \tilde{\mathbf{A}}_* \tilde{\mathbf{D}}^{-\frac{1}{2}} \mathbf{Z}_*^{(m)} \mathbf{W}_*^{(m)}\right) \qquad (3)$$

where $\tilde{\mathbf{A}}_l = \frac{1}{2}\left(\tilde{\mathbf{A}} + \tilde{\mathbf{D}}\right)$ for low-pass and $\tilde{\mathbf{A}}_h = \frac{1}{2}\left(\tilde{\mathbf{D}} - \tilde{\mathbf{A}}\right)$ for high-pass spectral encoding.

The **low-pass graph filter** is characterized by $\tilde{\mathbf{A}}_{l,\text{sym}} = \tilde{\mathbf{D}}^{-1/2} \tilde{\mathbf{A}}_l \tilde{\mathbf{D}}^{-1/2} = \mathbf{I} - \frac{1}{2}\tilde{\mathbf{L}}_{\text{sym}} = \mathbf{U}\left(\mathbf{I} - \frac{1}{2}\tilde{\mathbf{\Lambda}}\right)\mathbf{U}^\top$. This formulation reveals that $\tilde{\mathbf{A}}_{l,\text{sym}}$ exhibits a frequency response $p_l(\tilde{\lambda}_i) = 1 - \frac{1}{2}\tilde{\lambda}_i$. The response function is *monotonically* decreasing over $\tilde{\lambda}_i \in [0, 2]$, thereby attenuating high-frequency components while preserving smooth graph signals. This enables capturing local structural patterns and maintaining graph regularity in embeddings. Similarly, the **high-pass graph filter** is defined by $\tilde{\mathbf{A}}_{h,\text{sym}} = \tilde{\mathbf{D}}^{-1/2} \tilde{\mathbf{A}}_h \tilde{\mathbf{D}}^{-1/2} = \frac{1}{2}\tilde{\mathbf{L}}_{\text{sym}} = \mathbf{U}\left(\frac{1}{2}\tilde{\mathbf{\Lambda}}\right)\mathbf{U}^\top$. This formulation indicates that $\tilde{\mathbf{A}}_{h,\text{sym}}$ acts as a spectral filter with frequency response $p_h(\tilde{\lambda}_i) = \frac{1}{2}\tilde{\lambda}_i$ (see (Wang et al., 2022a)).

Figure 3 illustrates the frequency response curves of all three

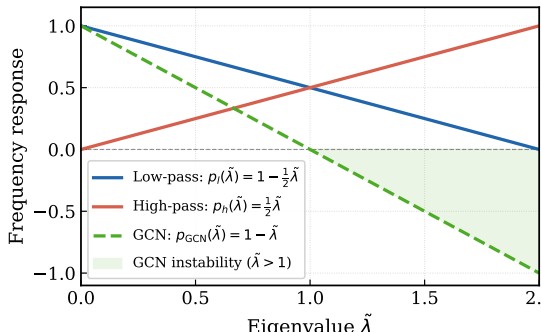

*Figure 3.* Spectral frequency response of graph filters

filters over $\tilde{\lambda} \in [0, 2]$. The low-pass branch $p_l(\tilde{\lambda}) = 1 - \frac{1}{2}\tilde{\lambda}$ is strictly monotonically decreasing, smoothly attenuating high-frequency components while fully preserving smooth, structure-aware signals at $\tilde{\lambda} \approx 0$. The high-pass branch $p_h(\tilde{\lambda}) = \frac{1}{2}\tilde{\lambda}$ is strictly monotonically increasing, suppressing low-frequency signals while amplifying fine-grained discriminative variations at $\tilde{\lambda} \approx 2$. In contrast, the standard GCN filter $p_{\text{GCN}}(\tilde{\lambda}) = 1 - \tilde{\lambda}$ becomes negative for $\tilde{\lambda} > 1$, introducing instability. Appendix E empirically validates these analytical properties.

**Theorem 3.2** (Discriminativity of dual-pass GCN encoder). *Let $\mathbf{z}_i^{\text{low}} \in \mathbb{R}^{d_1}$ and $\mathbf{z}_i^{\text{high}} \in \mathbb{R}^{d_2}$ denote node embeddings from low-pass and high-pass GCN encoders, respectively, and* dual-pass embedding *is defined as the concatenation $\mathbf{z}_i = [\mathbf{z}_i^{\text{low}} \, \| \, \mathbf{z}_i^{\text{high}}] \in \mathbb{R}^{d_1 + d_2}$. Using this architecture for both graphs $\mathcal{G}_1$ and $\mathcal{G}_2$, the dual-pass GCN encoder provides ideal node embeddings for graph alignment by satisfying:*

1. *Spectral locality preservation: the embedding $\mathbf{z}_i$ preserves neighborhood similarity comparably to $\mathbf{z}_i^{\text{low}}$.*

2. *Enhanced node discriminability: the embedding $\mathbf{z}_i$ provides superior node correspondence discrimination compared to either component alone.*

The proof is provided in Appendix B.

**Feature space filtering.** The dual-pass encoder operates in the feature space, combining low-pass filtering (for structural context) and high-pass filtering (for node distinctiveness) to produce embeddings $\mathbf{Z} = [\mathbf{Z}_l \, \| \, \mathbf{Z}_h]$ that are both structure-aware and highly discriminative. This addresses the first challenge of node distinctiveness degradation. Critically, this feature-level filtering is complemented by map-space filtering through our functional map module, which we describe next.

### 3.3. Latent space communication

While each GAE independently produces a latent space for its respective graph, resulting in misaligned embeddings,

we address this limitation by incorporating deep functional maps to learn explicit mappings between latent representations. Crucially, this module operates in the *space of maps* rather than the space of features. Rather than directly comparing raw embeddings, which may differ by arbitrary isometric transformations, we learn functional maps $\mathbf{C}^{12}$ and $\mathbf{C}^{21}$ that transform functions between latent spaces. These maps are optimized within our network using Equation 2, where $\mathbf{F}_1$ and $\mathbf{F}_2$ represent embeddings from the shared dual-pass encoder.

To facilitate latent space communication, our framework leverages spectral geometry principles and a regularized functional map module that enforces structural constraints. The *bijectivity* and *orthogonality* regularizations act as a low-pass filter on correspondences, enforcing smoothness and robustness as a structural prior in map space. The bijectivity loss promotes invertibility by ensuring functions mapped between latent spaces and back are accurately reconstructed, enforcing structural consistency and mutual alignment. Formally, it is defined as:

$$\mathcal{L}_{\text{bij}} = \|\mathbf{C}_{12}\mathbf{C}_{21} - \mathbf{I}\|_F^2 + \|\mathbf{C}_{21}\mathbf{C}_{12} - \mathbf{I}\|_F^2. \quad (4)$$

The bijectivity constraint operates in the spectral domain and is therefore fully compatible with graphs of different sizes. Bijectivity here enforces invertibility of the linear operator between function spaces, not one-to-one node matching, preventing degenerate map solutions and encouraging geometric consistency between latent spaces (see (Roufosse et al., 2019; Attaiki et al., 2021)).

The orthogonality loss enforces that functional maps behave as partial isometries, preserving local geometry and structural information during cross-space transformations. This loss is given by:

$$\mathcal{L}_{\text{orth}} = \|\mathbf{C}_{12}\mathbf{C}_{12}^\top - \mathbf{I}\|_F^2 + \|\mathbf{C}_{21}^\top\mathbf{C}_{21} - \mathbf{I}\|_F^2. \quad (5)$$

These regularizations enable *geometry-aware alignment* of latent spaces, facilitating reliable cross-graph alignment without requiring any ground-truth correspondences and effectively bridging independently learned embeddings.

Together, the dual-pass encoder and functional map module form a unified framework that addresses the dual challenges of graph alignment. The dual-pass encoder injects high-frequency discriminability into the feature space (addressing node distinctiveness), and the functional map module aligns latent spaces via bijective and isometric transformations while acting as a low-pass filter on correspondences. Importantly, the functional map module contributes to alignment *indirectly*: its losses and regularizations act as geometric constraints that guide the encoder to produce mutually alignable embeddings during training.

## 3.4. Graph decoder

Given latent node embeddings $\mathbf{Z} = f_\theta(\mathbf{X}, \mathbf{A}) \in \mathbb{R}^{|V| \times d}$ produced by the encoder, the decoder reconstructs graph structure using inner product operations $\hat{\mathbf{A}}_s = \sigma(\mathbf{Z}_s\mathbf{Z}_s^\top)$, and $\hat{\mathbf{A}}_t = \sigma(\mathbf{Z}_t\mathbf{Z}_t^\top)$, where $\sigma(\cdot)$ denotes the element-wise sigmoid function.

## 3.5. Model optimization and training Loss

Our model jointly optimizes GAE parameters and functional maps $\mathbf{C}_{12}$, $\mathbf{C}_{21}$ through end-to-end training. Given embeddings $\mathbf{Z}_1 = f_\theta(\mathbf{X}_s, \mathbf{A}_s)$ and $\mathbf{Z}_2 = f_\theta(\mathbf{X}_t, \mathbf{A}_t)$, we project them into spectral domains using graph Laplacian eigenvectors, yielding descriptors $\hat{\mathbf{F}}_1$ and $\hat{\mathbf{F}}_2$. Functional maps $\mathbf{C}_{12} \in \mathbb{R}^{k \times k}$ and $\mathbf{C}_{21} \in \mathbb{R}^{k \times k}$ align these spectral features via:

$$\mathcal{L}_{\text{FM}}^{12} = \alpha \left\|\mathbf{C}_{12}\hat{\mathbf{F}}_1 - \hat{\mathbf{F}}_2\right\|_F^2 + \beta \|\Lambda_2\mathbf{C}_{12} - \mathbf{C}_{12}\Lambda_1\|_F^2 \quad (6)$$

$$\mathcal{L}_{\text{FM}}^{21} = \alpha \left\|\mathbf{C}_{21}\hat{\mathbf{F}}_2 - \hat{\mathbf{F}}_1\right\|_F^2 + \beta \|\Lambda_1\mathbf{C}_{21} - \mathbf{C}_{21}\Lambda_2\|_F^2 \quad (7)$$

We incorporate these objectives as differentiable loss terms, with $\mathbf{C}_{12}$ and $\mathbf{C}_{21}$ as trainable parameters optimized end-to-end via backpropagation. These losses are combined with the standard GAE reconstruction loss, minimizing binary cross-entropy between the observed adjacency matrix $\mathbf{A}$ and its reconstruction $\hat{\mathbf{A}}$:

$$\mathcal{L}_{\text{rec}} = \text{BCE}(\mathbf{A}_s, \hat{\mathbf{A}}_s) + \text{BCE}(\mathbf{A}_t, \hat{\mathbf{A}}_t), \quad (8)$$

where $\text{BCE}(\cdot, \cdot)$ denotes the element-wise binary cross-entropy loss. The overall training objective combines the training loss and regularization terms in a weighted sum:

$$\mathcal{L}_{\text{total}} = \mathcal{L}_{\text{rec}} + \lambda_{\text{FM}} \left(\mathcal{L}_{\text{FM}}^{12} + \mathcal{L}_{\text{FM}}^{21}\right) + \lambda_{\text{bij}} \mathcal{L}_{\text{bij}} + \lambda_{\text{orth}} \mathcal{L}_{\text{orth}} \quad (9)$$

The entire architecture is trained end-to-end via gradient descent, ensuring that functional maps and embeddings co-evolve to produce structure-aware cross-graph correspondences.

## 3.6. Node alignment

Given learned embeddings, we compute the cosine similarity matrix $\mathbf{S} = \hat{\mathbf{Z}}_s\hat{\mathbf{Z}}_t^\top$ between $\ell_2$-normalized node embeddings $\hat{\mathbf{Z}}_s, \hat{\mathbf{Z}}_t \in \mathbb{R}^{N \times d}$ from source and target graphs. Node correspondences are predicted using greedy matching, iteratively selecting the highest similarity unmatched pairs until complete one-to-one alignment is achieved.

## 4. Experiments

In this section, we aim to address the following research questions: (1) **robustness:** is GADL more robust to feature

and structural inconsistencies than existing state-of-the-art graph alignment methods? (2) **effectiveness:** does GADL outperform state-of-the-art methods on real-world graph alignment tasks? (3) **latent space alignment:** how well does the functional map module qualitatively align node embeddings in latent space? (4) **generalization:** how effectively does GADL generalize to vision-language alignment? (5) **ablation analysis:** what is the contribution of each component in GADL to the overall alignment performance? (6) **encoder evaluation:** how does the proposed *dual-pass GCN encoder* improve node embeddings over standard GNNs? (7) **node discriminability:** does the dual-pass encoder preserve node discriminability? (8) **hyperparameter sensitivity:** how sensitive is GADL to hyperparameter variations?

A comprehensive description of the experimental setup, including benchmarks, baselines, evaluation metrics, and experimental settings, is provided in the Appendix C.

### 4.1. Robustness: evaluation on semi-synthetic benchmarks

To evaluate the robustness of the proposed GADL model under structural inconsistencies, we conduct experiments on six semi-synthetic benchmark datasets following (He et al., 2025), generating perturbed graph pairs with perturbation levels of 0%, 1%, and 5% (setup in Appendix C.3).

Table 1 compares GADL against state-of-the-arts: Netsimile (Berlingerio et al., 2013), Final (Zhang & Tong, 2016), GAlign (Trung et al., 2020), WAlign (Gao et al., 2021a), GAE (Kipf & Welling, 2016), T-GAE(He et al., 2025), and SLOTAlign (Tang et al., 2023a). Results show mean matching accuracy and standard deviation across 10 randomly generated target graphs under structural perturbations of 0%, 1%, and 5%. Results for Final, WAlign, and GAE are from He et al. (2025), while GAlign and SLOTAlign are reproduced. Entries marked "–" indicate scalability failures.

The results yield several key observations: (1) GADL consistently ranks among the top performers across datasets, maintaining high accuracy even with 5% perturbations while baselines show sharp degradation under structural noise. (2) Embedding-based methods (T-GAE, GAlign, GADL) generally outperform optimal-transport-based methods (Final). SLOTAlign, combining learning and optimization, achieves competitive but suboptimal results compared to pure learning-based models.

A notable observation is the dramatic performance degradation of standard GAE under structural perturbations. This phenomenon directly manifests the latent space misalignment problem illustrated in Figure 1 (c): without explicit alignment objectives or geometric constraints, GAE tends to produce embeddings that are highly sensitive to structural noise. T-GAE partially mitigates this issue through

transferable pre-training on graph families, which improves generalization to structural variations. However, it still lacks explicit geometric constraints to consistently align latent spaces. GADL incorporates these geometric constraints and achieves stronger robustness through two mechanisms: a dual-pass encoder that preserves discriminative node features and a geometry-aware functional map module that explicitly enforces geometric consistency between latent spaces.

### 4.2. Effectiveness: evaluation on real-world benchmarks

We evaluate the effectiveness of the proposed GADL method on two real-world noisy graph datasets with partial node alignment: Douban Online-Offline and ACM-DBLP. These benchmarks involve distinct graphs with partially aligned nodes. Performance is measured using Hit@$k$, the proportion of ground-truth nodes ranked in the top-$k$ predictions. Results are reported in Table 2.

The results reveal key insights: (1) GADL consistently achieves the highest alignment accuracy, outperforming all baselines, outperforming all baselines with substantial margins over second-best models (T-GAE on ACM-DBLP, SLOTAlign on Douban). (2) Compared to T-GAE, which employs a GIN encoder but lacks latent-space communication, our GADL model demonstrates superior performance through its dual-pass GCN encoder architecture integrated with latent-space communication. (3) Learning-based methods (T-GAE, GADL) outperform optimal-transport approaches (SLOTAlign) on larger benchmarks, demonstrating better robustness to structural variations that violate optimal transport assumptions.

Additional experiments on ablation analysis, encoder evaluation, node discriminability, and hyperparameter sensitivity are provided in Appendices D, F, G, and J, respectively. An analysis of oversmoothing behavior across GNN depth is provided in Appendix H. A detailed computational complexity and runtime analysis is presented in Appendix I.

### 4.3. Latent space alignment: qualitative evaluation of node embedding

Figure 4 visualizes t-SNE projections of the learned node embeddings on the Douban Online-Offline dataset, comparing embeddings obtained without and with the functional map module.

Each colored circle and triangle of the same color represent a ground-truth corresponding node pair, connected by a line (50 pairs are shown in each figure). In the left panel (without alignment), connecting lines are long and point in inconsistent directions, indicating that corresponding nodes are mapped to geometrically unrelated regions of their respective latent spaces. In the right panel (with alignment),

*Table 1.* Robustness evaluation under different structural inconsistency levels (%).

| Dataset | Perturb. | NetSimile | Final | GAlign | WAlign | GAE | T-GAE | SLOTAlign | GADL |
|---------|----------|-----------|-------|--------|--------|-----|-------|-----------|------|
| Celegans | 0% | 72.7 ± 0.9 | 92.2 ± 1.2 | 81.67 ± 0.7 | 88.4 ± 1.6 | 86.3 ± 1.3 | 91.0 ± 1.1 | 91.12 ± 0.2 | **92.82 ± 0.9** |
| | 1% | 66.3 ± 3.8 | 33.2 ± 7.8 | 66.23 ± 0.8 | 80.7 ± 3.0 | 33.2 ± 8.4 | 86.5 ± 1.1 | 85.25 ± 0.6 | **88.07 ± 0.7** |
| | 5% | 41.1 ± 13.0 | 10.4 ± 2.7 | 49.22 ± 1.6 | 42.4 ± 21.1 | 6.5 ± 2.4 | 69.2 ± 2.1 | 70.05 ± 0.4 | **71.74 ± 0.2** |
| Arena | 0% | 94.7 ± 0.3 | 97.5 ± 0.3 | 93.02 ± 0.4 | 97.4 ± 0.5 | 97.6 ± 0.4 | 97.8 ± 0.4 | 96.22 ± 0.5 | **98.27 ± 0.3** |
| | 1% | 87.8 ± 1.0 | 32.5 ± 5.9 | 87.46 ± 0.6 | 90.0 ± 3.1 | 30.1 ± 17.6 | 96.0 ± 1.0 | 95.24 ± 0.4 | **96.86 ± 0.4** |
| | 5% | 52.3 ± 5.3 | 7.2 ± 2.6 | 64.96 ± 1.2 | 30.4 ± 17.5 | 1.4 ± 1.4 | 78.6 ± 2.5 | 78.5 ± 0.6 | **80.69 ± 0.4** |
| Douban | 0% | 46.4 ± 0.4 | 89.9 ± 0.3 | 56.50 ± 1.4 | 90.0 ± 0.4 | 89.5 ± 0.4 | 90.1 ± 0.3 | 88.17 ± 0.3 | **90.71 ± 0.4** |
| | 1% | 40.0 ± 1.2 | 27.8 ± 5.7 | 51.40 ± 0.5 | 77.2 ± 4.8 | 38.3 ± 16.4 | 87.3 ± 0.4 | 85.83 ± 1.2 | **87.94 ± 0.1** |
| | 5% | 20.7 ± 4.6 | 7.8 ± 3.0 | 29.97 ± 2.2 | 36.6 ± 13.4 | 0.6 ± 0.3 | **70.2 ± 2.5** | 67.42 ± 0.5 | 69.62 ± 0.1 |
| Cora | 0% | 73.7 ± 0.4 | 87.5 ± 0.7 | 74.15 ± 0.7 | 87.2 ± 0.4 | 87.1 ± 0.8 | 87.5 ± 0.4 | 87.74 ± 0.6 | **88.62 ± 0.3** |
| | 1% | 66.4 ± 1.6 | 30.0 ± 3.3 | 68.53 ± 0.4 | 80.1 ± 1.2 | 57.9 ± 5.3 | 85.1 ± 0.5 | 84.66 ± 0.1 | **85.78 ± 0.1** |
| | 5% | 41.2 ± 3.3 | 6.7 ± 2.8 | 45.67 ± 0.8 | 33.4 ± 7.3 | 9.6 ± 2.7 | 67.7 ± 1.3 | 67.8 ± 0.3 | **71.12 ± 0.4** |
| DBLP | 0% | 63.7 ± 0.2 | 85.6 ± 0.2 | 66.43 ± 0.6 | 85.6 ± 0.2 | 85.2 ± 0.3 | 85.6 ± 0.2 | – | **85.82 ± 0.0** |
| | 1% | 55.1 ± 1.7 | 15.2 ± 3.3 | 59.00 ± 0.5 | 73.1 ± 1.6 | 19.4 ± 0.6 | **83.3 ± 0.4** | – | 82.77 ± 0.3 |
| | 5% | 19.5 ± 4.8 | 2.7 ± 0.9 | 38.84 ± 0.2 | 15.9 ± 8.3 | 1.4 ± 0.2 | 60.8 ± 1.9 | – | **62.49 ± 0.3** |
| Coauthor CS | 0% | 90.9 ± 0.1 | 97.6 ± 0.1 | 92.18 ± 1.5 | 97.5 ± 0.2 | 97.6 ± 0.3 | 97.6 ± 0.1 | – | **97.76 ± 0.1** |
| | 1% | 75.2 ± 2.2 | 13.3 ± 5.0 | 81.15 ± 0.7 | 75.2 ± 5.4 | 49.5 ± 7.8 | 93.2 ± 0.8 | – | **94.11 ± 0.4** |
| | 5% | 26.3 ± 6.0 | 2.0 ± 0.4 | 30.41 ± 0.1 | 11.3 ± 7.5 | 0.6 ± 0.1 | 66.0 ± 1.4 | – | **68.54 ± 1.2** |

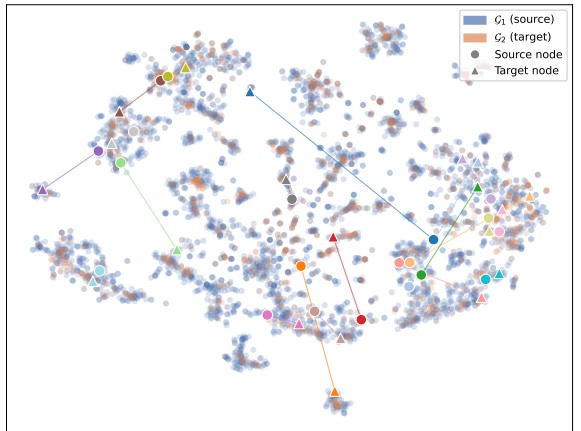
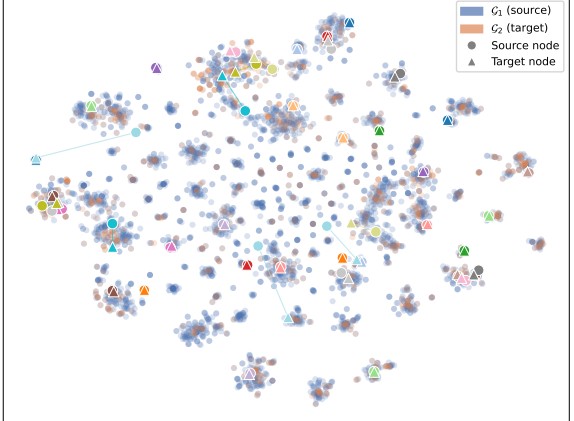

(a) Without Latent Space Alignment (Functional Map Module)      (b) With Latent Space Alignment (Functional Map Module)

*Figure 4.* t-SNE visualization of learned node embeddings on the Douban Online-Offline dataset.

lines are substantially shorter and corresponding pairs land in much closer proximity, demonstrating that the functional map module successfully reduces cross-graph latent space misalignment.

More broadly, with latent space alignment enabled, corresponding nodes tend to be either co-located within similar clusters or occupying consistent positions across clusters, indicating a more coherent geometric structure between the two graphs. Without latent space alignment, this consistency is weaker and pairs appear scattered across unrelated regions. Quantitatively, this is confirmed by the ablation study in Table 6, where removing the functional map module (GADL w/o latent-space alignment) causes a drop of 3.96 points in Hit@1 on Douban Online-Offline, the dataset with the largest structural heterogeneity.

### 4.4. Generalization: evaluation on vision–language benchmarks

Latent space alignment is a special case of graph alignment, relying only on embeddings without explicit structure. To highlight this generality, we further evaluate our method on vision-language alignment benchmarks, where the task involves aligning latent representations from diverse pretrained vision and language models. We evaluate latent space alignment across multiple benchmarks using representations from diverse pretrained vision and language models. Full experimental details are provided in Appendix C.5.

Table 3 summarizes the vision-language alignment accuracies on four datasets using three pretrained vision models (CLIP (Ramesh et al., 2022), DeiT (Touvron et al., 2021), and DINOv2 (Oquab et al., 2023)) and two pre-

*Table 2.* Performance of graph alignment methods on real-world benchmarks.

| Method | ACM-DBLP | | | | Douban Online-Offline | | | |
|---|---|---|---|---|---|---|---|---|
| | Hit@1 | Hit@5 | Hit@10 | Hit@50 | Hit@1 | Hit@5 | Hit@10 | Hit@50 |
| NetSimile | 2.59 | 8.32 | 12.09 | 26.42 | 1.07 | 2.77 | 4.74 | 15.03 |
| GAE | 8.10 | 22.50 | 30.10 | 45.10 | 3.30 | 9.20 | 14.10 | 32.10 |
| GAlign | 73.26 | 91.24 | 95.09 | 98.37 | 41.32 | 62.43 | 71.37 | 87.65 |
| WAlign | 62.02 | 81.96 | 87.31 | 93.89 | 36.40 | 53.94 | 67.08 | 85.33 |
| T-GAE | 73.89 | 91.73 | 95.33 | 98.22 | 36.94 | 60.64 | 69.77 | 88.62 |
| SLOTAlign | 66.04 | 84.06 | 87.95 | 94.65 | 51.43 | 53.43 | 77.73 | 90.23 |
| GADL | **88.63** | **94.76** | **96.16** | **98.41** | **53.31** | **73.61** | **80.67** | **94.18** |

*Table 3.* Vision-language alignment across four datasets using three pretrained vision models (CLIP, DeiT, and DINOv2) and two pretrained language models: Lan. model 1 (all-mpnet-base-v2) and Lan. model 2 (all-roberta-large-v1).

| Method | CIFAR-10 | | CINIC-10 | | CIFAR-100 | | ImageNet-100 | |
|---|---|---|---|---|---|---|---|---|
| | Lan. model 1 | Lan. model 2 | Lan. model 1 | Lan. model 2 | Lan. model 1 | Lan. model 2 | Lan. model 1 | Lan. model 2 |
| | | | | **CLIP** - (ViT-L/14@336) | | | | |
| LocalCKA | 25.0 ± 10.5 | 17.0 ± 15.9 | 30.0 ± 0.0 | 4.0 ± 5.0 | **24.00 ± 1.41** | 13.67 ± 0.47 | 8.00 ± 1.41 | 8.33 ± 0.47 |
| OT | 0.0 ± 0.0 | 10.0 ± 0.0 | 49.5 ± 2.2 | 2.0 ± 4.1 | 1.00 ± 0.00 | 1.67 ± 0.47 | 1.33 ± 0.47 | 1.00 ± 0.00 |
| FAQ | 12.0 ± 10.1 | 0.5 ± 2.2 | 30.5 ± 2.2 | 0.0 ± 0.0 | 2.33 ± 1.25 | 2.67 ± 1.70 | 4.33 ± 1.70 | 2.33 ± 1.25 |
| MPOpt | 0.0 ± 0.0 | 0.0 ± 0.0 | 0.0 ± 0.0 | 0.0 ± 0.0 | 1.67 ± 1.25 | 2.67 ± 1.25 | 4.67 ± 2.05 | 2.67 ± 0.47 |
| Gurobi | 20.5 ± 6.0 | 47.0 ± 7.3 | 50.0 ± 0.0 | 80.0 ± 0.0 | 2.11 ± 1.29 | 3.44 ± 1.27 | 3.22 ± 2.35 | 4.50 ± 1.50 |
| Hahn-Grant | 25.0 ± 10.5 | 47.0 ± 7.3 | 50.0 ± 0.0 | **80.0 ± 0.0** | 2.33 ± 1.25 | 3.00 ± 2.16 | 4.93 ± 2.05 | 4.67 ± 1.70 |
| **GADL** | **32.0 ± 4.0** | **48.0 ± 5.20** | **56.7 ± 4.8** | **80.0 ± 0.0** | 16.8 ± 4.1 | **14.2 ± 5.9** | **19.6 ± 8.3** | **10.4 ± 5.7** |
| | | | | **DeiT** - (DeiT-B/16d@384) | | | | |
| LocalCKA | 24.0 ± 9.9 | 20.0 ± 5.6 | **68.0 ± 8.9** | 0.0 ± 0.0 | 10.33 ± 0.94 | **23.33 ± 0.47** | 8.33 ± 1.70 | 9.33 ± 0.47 |
| OT | 12.0 ± 4.1 | 10.0 ± 0.0 | 20.0 ± 0.0 | 0.0 ± 0.0 | 2.33 ± 0.94 | 1.67 ± 0.47 | 2.00 ± 0.00 | 0.67 ± 0.47 |
| FAQ | 40.0 ± 15.2 | 22.5 ± 9.7 | 55.5 ± 5.1 | 0.0 ± 0.0 | 4.33 ± 0.47 | 1.33 ± 1.25 | 3.67 ± 0.47 | 3.33 ± 0.94 |
| MPOpt | 0.0 ± 0.0 | 0.0 ± 0.0 | 0.0 ± 0.0 | 0.0 ± 0.0 | 0.33 ± 0.47 | 0.67 ± 0.94 | 2.67 ± 2.36 | 1.00 ± 0.82 |
| Gurobi | 28.5 ± 3.7 | 59.0 ± 3.1 | 10.0 ± 0.0 | 40.0 ± 0.0 | 3.67 ± 2.49 | 3.11 ± 1.91 | 3.56 ± 1.57 | 3.00 ± 1.00 |
| Hahn-Grant | 28.5 ± 3.7 | **59.0 ± 3.1** | 10.0 ± 0.0 | 40.0 ± 0.0 | 1.33 ± 0.47 | 5.33 ± 1.25 | 1.67 ± 2.36 | 1.33 ± 1.25 |
| **GADL** | **44.0 ± 4.9** | 52.00 ± 7.2 | 60.0 ± 8.9 | **46.00 ± 4.9** | **10.8 ± 4.7** | 13.4 ± 4.8 | **20.2 ± 8.6** | **13.0 ± 6.8** |
| | | | | **DINOv2** - (ViT-G/14) | | | | |
| LocalCKA | 37.5 ± 28.8 | 18.5 ± 29.2 | 52.5 ± 31.1 | 57.0 ± 13.4 | 4.00 ± 0.82 | 4.67 ± 0.94 | 5.33 ± 0.47 | **6.00 ± 0.82** |
| OT | 30.0 ± 13.8 | 33.5 ± 19.8 | 77.5 ± 6.4 | 15.5 ± 7.6 | 1.00 ± 0.00 | 1.00 ± 0.00 | 1.00 ± 0.00 | 0.33 ± 0.47 |
| FAQ | 37.5 ± 21.2 | 38.0 ± 29.8 | 31.0 ± 4.5 | 29.5 ± 2.2 | 4.33 ± 0.94 | 3.33 ± 1.25 | 3.00 ± 0.82 | 4.33 ± 2.05 |
| MPOpt | 73.5 ± 17.9 | 94.0 ± 18.5 | 79.0 ± 3.1 | 47.0 ± 46.0 | 1.33 ± 1.25 | 0.33 ± 0.47 | 4.00 ± 0.82 | 0.67 ± 0.47 |
| Gurobi | 69.5 ± 24.2 | **100.0 ± 0.0** | 79.0 ± 3.1 | **100.0 ± 0.0** | 2.56 ± 1.64 | 1.78 ± 1.31 | 1.50 ± 0.50 | 2.50 ± 0.50 |
| Hahn-Grant | 69.5 ± 24.2 | **100.0 ± 0.0** | 79.0 ± 3.1 | **100.0 ± 0.0** | 4.00 ± 0.82 | 2.00 ± 1.41 | 6.33 ± 0.47 | 1.22 ± 0.92 |
| **GADL** | **100.0 ± 0.0** | **100.0 ± 0.0** | **100.0 ± 0.0** | 42.0 ± 11.6 | **20.8 ± 5.2** | **19.6 ± 6.2** | **8.8 ± 4.2** | 5.8 ± 1.3 |

trained language models from SentenceTransformers library (Reimers & Gurevych, 2019) (`all-mpnet-base-v2` and `all-roberta-large-v1`). Comprehensive results are in the Appendix K. Results on CIFAR-100 and ImageNet-100 are reproduced using official implementations.

The results highlight key insights. (1) GADL outperforms most baselines, demonstrating substantial benefits beyond optimization and optimal transport frameworks. (2) Most baselines achieve near-chance accuracies ($\leq 10\%$) with occasional inconsistent successes, even sophisticated solvers like Gurobi fail in certain settings. Performance gaps with GADL become pronounced on challenging benchmarks (CIFAR-100, ImageNet-100), highlighting limitations of treating alignment as pure assignment optimization. (3) Pretrained model choice critically impacts performance, while DINOv2 and DeiT excel on smaller datasets, CLIP consistently outperforms on larger benchmarks.

## 5. Conclusion

We present GADL, a novel framework for unsupervised graph alignment that combines dual-pass encoding with geometry-aware latent space communication. Comprehensive experiments demonstrate consistent performance gains across diverse benchmarks, with successful application to vision-language tasks validating the broader utility of the framework beyond traditional graph domains. While promising, our approach incurs modest computational overhead from dual-pass encoding compared to standard GCN and requires careful hyperparameter tuning. Additionally, a key limitation is the lack of large-scale benchmarks with reliable ground truth for graph alignment, which remains an important direction for future work. Future work will also focus on adaptive spectral filtering and efficient embedding strategies, with potential extensions to molecular networks, social graphs, and multi-modal alignment tasks, including a more thorough evaluation on vision-language benchmarks.

## Acknowledgements

We would like to thank the anonymous reviewers for their constructive feedback and valuable suggestions. Parts of this work were supported by the ERC Consolidator Grant 101087347 (VEGA), as well as gifts from Ansys Inc., and Adobe Research.

## Impact Statement

This paper contributes to the advancement of graph representation learning by enabling reliable cross-graph node matching without ground-truth correspondences. While our work holds potential societal consequences common to graph learning methods, no direct negative implications need to be highlighted at this time.

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

# A. Related work

Extensive research has addressed the graph alignment problem, with existing methods broadly categorized into three families based on their alignment strategies: optimization-based, optimal transport-based, and embedding-based approaches. These methods also differ in terms of the level of supervision required, ranging from unsupervised techniques to semi-supervised methods (which rely on partially paired nodes), and fully supervised approaches.

Traditional graph alignment methods formulate the problem as an optimization task, typically as a Quadratic Assignment Problem (QAP), seeking node permutations that minimize discrepancies between source and target adjacency matrices. IsoRank (Singh et al., 2008) represents a seminal approach, employing a PageRank-inspired algorithm to compute node similarity matrices based on neighbor similarity for unsupervised alignment. BigAlign (Koutra et al., 2013) extends this framework by incorporating both structural and attribute information to enhance alignment accuracy. FINAL (Zhang & Tong, 2016) addresses scalability through matrix factorization, combining global structural consistency with partial anchor constraints.

These optimization-based approaches often struggle with scalability due to the NP-hard nature of QAP, though approximation strategies and relaxations can make them tractable on medium-sized networks. While primarily unsupervised, they can accommodate semi-supervised settings by incorporating known anchor pairs as hard or soft constraint

Optimal transport-based methods model each graph as a probability distribution over its nodes and seek a transport plan, i.e., a soft correspondence, that minimizes a divergence such as the Wasserstein or Gromov-Wasserstein distance between the distributions. This framework offers a principled approach to graph alignment by optimizing the transport cost between node distributions. Unlike the hard alignments produced by QAP-based methods, optimal transport typically yields soft alignment matrices, allowing for uncertainty and partial correspondences.

A notable early contribution in this category is WAlign (Gao et al., 2021a), which jointly learns node embeddings and alignments by minimizing Wasserstein distance between graphs in a shared embedding space using a lightweight GCN and Wasserstein distance discriminator. Building on this direction, FGW (Tang et al., 2023b) employs Fused Gromov-Wasserstein distance to jointly align structural and attribute information through a coarse-to-fine matching scheme. PARROT (Zeng et al., 2023) extends this idea by running Random Walk with Restart (RWR) on both individual graphs and their Cartesian product, capturing more nuanced structural correspondence. GALOPA (Wang et al., 2023) integrates a GNN encoder with a self-supervised OT loss, jointly learning features and transport plans. To improve scalability, Wasserstein Wormhole (Haviv et al., 2024) introduces a transformer-based autoencoder that maps distributions into a latent space where Euclidean distances approximate Wasserstein distances, enabling efficient, linear-time graph comparisons.

These methods are typically unsupervised and particularly effective for noisy or incomplete graphs due to their probabilistic formulation and global alignment perspective.

Embedding-based methods learn vector representations for nodes in each graph and align them based on embedding similarity. This approach typically involves generating node embeddings, either independently or jointly, followed by alignment through nearest-neighbor search or learned mapping functions.

NetSimile (Berlingerio et al., 2013) represents an early embedding-based approach that uses handcrafted structural features (degree, clustering coefficient) to represent nodes and aligns graphs through direct feature vector comparison using similarity measures. GAlign (Trung et al., 2020) adopts an unsupervised approach where both graphs are independently encoded using a shared Graph Convolutional Network, with node embeddings aligned by minimizing distributional discrepancies such as Wasserstein distance between embedding spaces. NeXtAlign (Zhang et al., 2021) enhances representation learning through a cross-graph attention mechanism that enables nodes in one graph to attend to features in the other. This produces alignment-aware embeddings and improves performance in semi-supervised settings with known anchor node pairs. REGAL (Heimann et al., 2018) generates compact node embeddings by extracting structural features like node degree and local neighborhoods, then aligns nodes across graphs by matching their embeddings based on distance, enabling efficient and scalable graph alignment. GINA (Wang et al., 2022b) addresses hierarchical alignment by projecting node embeddings from Euclidean to hyperbolic space, learning linear transformations between geometries using anchor nodes to better capture scale-free and hierarchical structures in social and biological networks.

A foundational approach to embedding-based graph learning is the Graph Autoencoder (GAE) and its probabilistic extension, the Variational Graph Autoencoder (VGAE) (Kipf & Welling, 2016). These models use a GCN encoder to generate latent node embeddings, which are then used to reconstruct the adjacency matrix via an inner product decoder. Although originally

designed for link prediction, GAEs have become a common backbone for alignment tasks due to their ability to capture global graph structure in an unsupervised manner. Expanding on this foundation, T-GAE (He et al., 2025) addresses scalability through a transferable graph autoencoder trained on small graph families that generalizes to large, unseen networks without fine-tuning. This design enables strong alignment performance while significantly reducing training time and computational overhead. However, typical embedding-based methods often become unstable when graphs differ significantly in structure. SLOTAlign (Tang et al., 2023a) is developed to tackle the structure and feature inconsistencies commonly found in these embedding-based graph alignment methods. It formulates alignment as an optimal transport problem on learned intra-graph similarity matrices, combining optimal transport with embedding-based approaches.

In embedding-based graph alignment methods, the uniqueness and discriminative power of learned embeddings play a critical role in alignment accuracy. Several spectral GNN models have been proposed to go beyond low-frequency information in graph convolutional networks, including GPR-GNN (Chien et al., 2021), BernNet (He et al., 2021), TFE-GNN (Duan et al., 2024), and FAGCN (Bo et al., 2021). These models primarily target single-graph tasks, such as node classification, and leverage high-frequency graph signals to mitigate over-smoothing. While effective for these purposes, they are not directly applicable to graph alignment, which requires embeddings that are learned in a fully unsupervised manner and robust to structural inconsistencies across graphs.

Our approach addresses these challenges through a fundamentally different design. We introduce a dual-pass encoder with explicit low-pass and high-pass branches whose outputs are preserved, concatenated, and learned in a fully unsupervised manner. Crucially, we pair this with a functional map module that operates in the space of maps rather than features, enforcing latent space alignment across graphs while acting as a structural prior for smooth and robust correspondences. This design enables embeddings that are simultaneously discriminative and aligned across graphs, capabilities not provided by prior spectral GNN methods that operate solely in the feature space.

## B. Proof of Theorem 3.2

*Proof.* **Preliminary Definitions:** Let GCN filter $\tilde{\mathbf{A}}_{\text{GCN,sym}} = \mathbf{I} - \tilde{\mathbf{L}}_{\text{sym}} = \mathbf{U}(\mathbf{I} - \tilde{\mathbf{\Lambda}})\mathbf{U}^\top$, where $\mathbf{U} = [\mathbf{u}_1, \ldots, \mathbf{u}_n]$ is the eigenbasis, $\hat{\mathbf{\Lambda}} = \text{diag}(\hat{\lambda}_1, \ldots, \hat{\lambda}_n)$ is the diagonal matrix of eigenvalues, and each eigenvalue satisfies $\hat{\lambda}_i \in [0, 2]$. The spectral representation of node features is: $\mathbf{X} = \sum_{k=1}^n \hat{\mathbf{X}}_k \mathbf{u}_k$, where $\hat{\mathbf{X}}_k = \mathbf{u}_k^\top \mathbf{X}$.

- The low-pass component captures the smooth, global structure of the graph. It aggregates information from neighbors, producing embeddings: $\mathbf{Z}_{\text{low}} = \sum_{k=1}^n p_{\text{low}}(\hat{\lambda}_k) \hat{\mathbf{X}}_k \mathbf{u}_k$, where $p_{\text{low}}(\hat{\lambda}_k) = 1 - \frac{1}{2}\hat{\lambda}_k$. This captures smoothed signals over the graph, node embeddings are averages of their neighbors.

- The high-pass component captures complementary, local variations and finer structural details, given by: $\mathbf{Z}_{\text{high}} = \sum_{k=1}^n p_{\text{high}}(\hat{\lambda}_k) \hat{\mathbf{X}}_k \mathbf{u}_k$, where $p_{\text{high}}(\hat{\lambda}_k) = \frac{1}{2}\hat{\lambda}_k$.

**Claim 1 (Neighborhood preservation).** The dual-pass embedding $\mathbf{z}_i$ preserves neighborhood similarity as effectively as the low-pass embedding $\mathbf{z}_i^{\text{low}}$.

**Proof of Claim 1.** The key insight underlying local consistency is that neighborhood similarity is primarily encoded in low-frequency spectral components, which capture smooth variations across connected nodes.

For the dual-pass embeddings of nodes $i$ and $j$, the cosine similarity can be written as:

$$\langle \mathbf{z}_i, \mathbf{z}_j \rangle = \langle \mathbf{z}_i^{\text{low}}, \mathbf{z}_j^{\text{low}} \rangle + \langle \mathbf{z}_i^{\text{high}}, \mathbf{z}_j^{\text{high}} \rangle + \langle \mathbf{z}_i^{\text{low}}, \mathbf{z}_j^{\text{high}} \rangle + \langle \mathbf{z}_i^{\text{high}}, \mathbf{z}_j^{\text{low}} \rangle.$$

The filters are designed to be spectrally complementary,

$$p_{\text{low}}(\hat{\lambda}_k) + p_{\text{high}}(\hat{\lambda}_k) = (1 - \frac{1}{2}\hat{\lambda}_k) + \frac{1}{2}\hat{\lambda}_k = 1, \quad \forall \hat{\lambda}_k \in [0, 2].$$

Due to their complementary spectral responses, the low-pass and high-pass components are approximately orthogonal. To see this, note that the low-pass filter $p_{\text{low}}(\hat{\lambda}_k) = 1 - \frac{1}{2}\hat{\lambda}_k$ is monotonically decreasing, achieving maximum response at $\hat{\lambda}_k = 0$ and minimum at $\hat{\lambda}_k = 2$. Conversely, the high-pass filter $p_{\text{high}}(\hat{\lambda}_k) = \frac{1}{2}\hat{\lambda}_k$ is monotonically increasing, with

minimum response at $\hat{\lambda}_k = 0$ and maximum at $\hat{\lambda}_k = 2$. The spectral overlap between components is measured by the product $p_{\text{low}}(\hat{\lambda}_k) \cdot p_{\text{high}}(\hat{\lambda}_k) = \frac{1}{4}\hat{\lambda}_k(2 - \hat{\lambda}_k)$, which is maximized only at the intermediate eigenvalue $\hat{\lambda}_k = 1$ and approaches zero at both extremes.

This spectral disjointness ensures that the high-pass component adds complementary discriminative information without interfering with the neighborhood-preserving properties encoded in the low-frequency domain by the low-pass component. Consequently, the dual-pass embedding $\mathbf{z}_i = [\mathbf{z}_i^{\text{low}} \| \mathbf{z}_i^{\text{high}}]$ preserves neighborhood similarity as effectively as the low-pass component $\mathbf{z}_i^{\text{low}}$ alone, since the neighborhood-relevant information is fully retained while additional discriminative power is gained.

**Claim 2 (Enhanced discriminability for node correspondence).** For node correspondence tasks, the dual-pass embedding $\mathbf{z}_i$ provides superior discriminability compared to either $\mathbf{z}_i^{\text{low}}$ or $\mathbf{z}_i^{\text{high}}$ alone. That is, false correspondences are less likely under similarity computed via $\mathbf{z}_i$.

**Proof of Claim 2.** Discriminability is measured by the separation margin between the distributions of similarities for *corresponding pairs* $\mathcal{C} = \{(i,j) : i \in \mathcal{V}_1, \ j \in \mathcal{V}_2, \ i \leftrightarrow j\}$ and *non-corresponding pairs* $\mathcal{N} = \{(i,j) : i \in \mathcal{V}_1, \ j \in \mathcal{V}_2, \ i \nleftrightarrow j\}$.

As we mentioned, the dual-pass filter design ensures perfect spectral complementarity: the low-pass and high-pass filters have anti-correlated frequency responses, ensuring that their contributions are nearly independent. Consequently, their mutual information is bounded, $I(\mathbf{z}_i^{\text{low}}; \mathbf{z}_i^{\text{high}}) \leq \epsilon$, for small $\epsilon > 0$, reflecting the opposing spectral emphasis.

For any $\ell_2$-induced metric, the squared distance between dual-pass embeddings decomposes as:

$$d(\mathbf{z}_i, \mathbf{z}_j)^2 = d(\mathbf{z}_i^{\text{low}}, \mathbf{z}_j^{\text{low}})^2 + d(\mathbf{z}_i^{\text{high}}, \mathbf{z}_j^{\text{high}})^2 + 2\langle \mathbf{z}_i^{\text{low}} - \mathbf{z}_j^{\text{low}}, \mathbf{z}_i^{\text{high}} - \mathbf{z}_j^{\text{high}} \rangle.$$

Under the approximate orthogonality condition, the cross-term is negligible, yielding:

$$d(\mathbf{z}_i, \mathbf{z}_j)^2 \approx d(\mathbf{z}_i^{\text{low}}, \mathbf{z}_j^{\text{low}})^2 + d(\mathbf{z}_i^{\text{high}}, \mathbf{z}_j^{\text{high}})^2.$$

The separation margin is thus defined as $\Delta = \min_{(i,j)\in\mathcal{C}} \text{sim}(\mathbf{z}_i, \mathbf{z}_j) - \max_{(i,k)\in\mathcal{N}} \text{sim}(\mathbf{z}_i, \mathbf{z}_k)$.

The critical observation is that the two components offer complementary discriminative power:

- **Case A (Low-pass insufficient):** When graphs share similar global structure but differ in local details, $\mathbf{z}_i^{\text{low}}$ may yield high similarity for non-corresponding pairs. However, $\mathbf{z}_i^{\text{high}}$ captures local differences, reducing false positives.

- **Case B (High-pass insufficient):** When local structures are noisy or similar, $\mathbf{z}_i^{\text{high}}$ may be unreliable. However, $\mathbf{z}_i^{\text{low}}$ provides stable global discrimination based on community structure and smooth attributes.

Together, these effects yield an additive improvement in discriminability. Formally, the dual-pass margin satisfies

$$\Delta_{\text{dual}} \geq \max(\Delta_{\text{low}}, \Delta_{\text{high}}) + \gamma,$$

where $\gamma > 0$ represents the additional discriminative contribution from orthogonal spectral information. This establishes that the dual-pass embedding provides superior node correspondence discrimination.

$\square$

# C. Experimental setup

This section describes the benchmarks, performance metrics, and experimental settings used for graph alignment evaluation.

## C.1. Benchmarks

Table 4 summarizes statistics for all experimental datasets. It includes six semi-synthetic graph alignment benchmarks consisting of graphs with varying sizes and properties to comprehensively evaluate the robustness of our approach. Additionally, two real-world graph alignment datasets with partial ground-truth node correspondences are included to assess overall performance. The datasets are as follows:

*Table 4.* Overview of datasets and their key properties

| Dataset | | #Nodes | #Edges | #Aligned Nodes | #Node Features | Description |
|---|---|---|---|---|---|---|
| Celegans | | 453 | 2025 | 453 | | Interactome |
| Arenas | | 1133 | 5451 | 1133 | | Email Communication |
| Douban | | 3906 | 7215 | 3906 | 7 | Social Network |
| Cora | | 2708 | 5278 | 2708 | | Citation Network |
| DBLP | | 17716 | 52867 | 17716 | | Citation Network |
| CoauthorCs | | 18333 | 81894 | 18333 | | Coauthor |
| ACM-DBLP | ACM | 9872 | 39561 | 6325 | 17 | Coauthor Network |
| | DBLP | 9916 | 44808 | | | Coauthor Network |
| Douban | Online | 3906 | 16328 | 1118 | 538 | Social Network |
| | Offline | 1118 | 3022 | | | Social Network |

- **Celegans:** This dataset models the protein-protein interaction network of *Caenorhabditis elegans*. Each node represents a protein, and edges indicate physical or functional interactions between proteins, making it useful for biological network analysis and alignment tasks involving molecular networks (Kunegis, 2013).

- **Arenas:** A communication network derived from email exchanges at the University Rovira i Virgili. Nodes correspond to individual users, and edges represent the presence of at least one email sent between them. It serves as a social interaction graph with temporal and communication patterns (Leskovec & Krevl, 2014).

- **Douban:** A social network from the Chinese movie review platform Douban, where nodes represent users, and edges capture friend or contact relationships. This dataset is commonly used to study social dynamics and network alignment in social media contexts (Zhang & Tong, 2016).

- **Cora:** A citation network of scientific papers where nodes are publications, and edges denote citation relationships. Cora is a benchmark dataset for graph mining and node classification, providing a structured academic citation graph ideal for evaluating graph-based learning models (Sen et al., 2008).

- **DBLP:** An extensive citation network aggregated from DBLP, Association for Computing Machinery (ACM), Microsoft Academic Graph (MAG), and other scholarly databases. It includes publication and citation information, widely used for testing graph alignment, clustering, and knowledge discovery tasks in academic networks (Pan et al., 2016).

- **CoauthorCs:** A co-authorship network in computer science that represents collaborations between authors. Nodes correspond to researchers, and edges indicate joint publications. It is often used to study community structure and author disambiguation in bibliographic databases (Sinha et al., 2015).

- **ACM-DBLP:** This dataset contains two co-authorship graphs from the ACM and DBLP databases. Nodes represent authors, and edges indicate co-authorship. Although collected independently, both graphs share overlapping authors, with 6,325 ground-truth alignments. Node features capture publication distributions across research venues. The dataset poses a challenging alignment task due to structural and feature discrepancies between the two graphs (Zhang & Tong, 2018).

- **Douban Online-Offline:** This dataset comprises two social graphs from the Douban platform, one based on online interactions and the other on offline event co-attendance. Both graphs share a subset of users, with 1,118 aligned nodes. Node features reflect user location distributions. The dataset is designed to evaluate alignment across heterogeneous and partially overlapping social networks (Zhang & Tong, 2016).

## C.2. Performance metrics

To assess graph alignment performance, we adopt two widely used metrics: alignment accuracy (Acc) and Hit@$k$. Alignment Accuracy (Acc) measures the proportion of correctly predicted node correspondences among all ground-truth aligned pairs, providing a direct measure of overall matching performance. Hit@$k$ evaluates whether the true corresponding node from the source graph appears within the top-$k$ predicted candidates for each node in the target graph. This metric reflects the ability of model to rank correct matches highly and is particularly useful for top-$k$ retrieval scenarios. Both metrics are computed using all available ground-truth node pairs, with higher values indicating better alignment quality.

## C.3. Experimental settings

In section 4.1, we follow the experimental setting introduced in the (He et al., 2025) for generating inconsistent graph pairs. Given a source graph $\mathcal{G}_s$ with adjacency matrix $\mathbf{A}$, we construct 10 perturbed and permuted target graphs using the transformation $\hat{\mathbf{A}} = \mathbf{P}(\mathbf{A} + \mathbf{M})\mathbf{P}^\top$, where $\mathbf{M} \in \{-1, 0, 1\}^{N \times N}$ introduces edge-level perturbations, and $\mathbf{P}$ is a random permutation matrix. The perturbation level is controlled by a parameter $p \in \{0, 1\%, 5\%\}$, representing the fraction of edges modified: $p|\mathcal{E}|$. We adopt seven structural node features from (Berlingerio et al., 2013): *node degree*, *clustering coefficient*, *average degree of neighbors*, *average clustering coefficient of neighbors*, *number of edges in ego-network*, *number of outgoing edges of ego-network*, and *number of neighbors of ego-network*. These descriptors provide compact, structure-aware representations for robust evaluation under varying structural inconsistencies.

All experiments are implemented using PyTorch 2.1.2 and PyTorch Geometric 2.5.0. Most benchmarks are run on servers equipped with NVIDIA A100 GPUs (CUDA 12.2), each providing 40 GB of memory. For large-scale datasets such as DBLP and Coauthor CS, we use NVIDIA H100 GPUs (CUDA 12.6) with 95 GB of memory, enabling efficient training and evaluation on high-complexity graphs. The implementation is available at `https://github.com/maysambehmanesh/GADL`.

## C.4. Hyperparameter selection

In the GADL framework, hyperparameters include: (1) architectural parameters: number of GCN layers, hidden dimensions, and spectral basis size ($k$); (2) weighting coefficients: $\lambda_{\text{FM}}$, $\lambda_{\text{bij}}$, $\lambda_{\text{orth}}$, $\alpha$, and $\beta$; and (3) optimization parameters: learning rate, and weight decay.

For the *architectural hyperparameter*, we select values guided by the structure of the graphs and computational considerations. For the number of GCN layers, we balance three factors: enabling sufficient long-range information propagation, computational efficiency, and avoiding oversmoothing. Small, dense graphs (Celegans, Arena, Cora) have short diameters and high connectivity, so 2 layers suffice to capture local neighborhoods while preserving node distinctiveness. Medium-sized, sparse graphs (Douban and Douban Online-Offline) require 5-6 layers because their lower connectivity demands deeper information propagation, and our dual-pass design supports this depth while still preserving discriminative embeddings. For large-scale graphs (ACM-DBLP, DBLP, CoauthorCS), we limit the depth to 2-3 layers: despite their size, these graphs are locally dense, and most useful structural information lies within 2-3 hops. Adding more layers increases computational cost without improving accuracy.

For hidden dimensions, we scale with problem complexity and richness of node features: 16 dimensions for semi-synthetic graphs with simple 7-dimensional structural features, 256 for Douban Online-Offline with its 538-dimensional *sparse* features, and 1024 for ACM-DBLP, which combines large scale with diverse structural patterns.

We set $k = 300$ for the spectral basis after testing a range of values and using principles from functional map theory. In functional map theory, increasing $k$ improves the ability of a linear functional map to approximate the underlying correspondence. If a valid node permutation exists, a sufficiently high-dimensional spectral basis can always represent it. In practice, we find that $k = 300$ provides stable alignment on our benchmarks while balancing accuracy and computational efficiency.

For the loss function weights $\lambda_{\text{FM}}$, $\lambda_{\text{bij}}$, $\lambda_{\text{orth}}$ and the functional map term weights $\alpha$, and $\beta$, we perform a *grid search* to systematically evaluate combinations of values. We set $\alpha = 10^{-3}$, $\beta = 10^{-2}$, $\lambda_{\text{FM}} = 1$, $\lambda_{\text{bij}} = 10^{-1}$, and $\lambda_{\text{orth}} = 10^{-1}$ for all benchmarks, though slight adjustments could improve results on individual datasets. A sensitivity analysis of loss function weights is presented in Appendix J.

The optimization hyperparameters, learning rate and weight decay, are set to standard values commonly used for Adam in graph learning tasks: a learning rate of $1e - 3$ and a weight decay of $5e - 4$. The model is trained end-to-end using the Adam optimizer based on the loss function in Equation 9.

For applying GADL to new problems, we recommend the following: (1) analyze graph properties such as diameter, sparsity, average degree, and clustering coefficient; (2) choose the number of layers based on graph structure, balancing long-range information propagation, computational efficiency, and avoiding oversmoothing; (3) scale hidden dimensions with graph size and feature richness; (4) select a sufficiently large spectral basis $k$ to reliably capture the underlying correspondence, while still balancing computational efficiency; and (5) perform a grid search to tune $\lambda_{\text{FM}}$, $\lambda_{\text{bij}}$, $\lambda_{\text{orth}}$, $\alpha$, and $\beta$.

## C.5. Details of vision–language experiment

### C.5.1. SETUP

We evaluate the vision-language alignment task on a range of benchmarks, including CIFAR-10 (Krizhevsky et al., 2009), CINIC-10 (Darlow et al., 2018), CIFAR-100 (Krizhevsky et al., 2009), and ImageNet-100 (Russakovsky et al., 2015), using representations extracted from diverse pretrained vision and language models. For each vision model, class-level representations are derived by averaging image-level embeddings within each class. Correspondingly, language representations are obtained by averaging embeddings generated from multiple textual prompts for each class. To enable application of our graph alignment method, we build a similarity graph from these representations, where each class-level embedding is treated as a node and connected to its $k$ most similar neighbors according to cosine similarity. Final node correspondences are obtained by computing cosine similarity between the aligned embeddings and applying the Hungarian algorithm for optimal assignment.

Note that baselines introduce variability across runs by randomly sampling half of the image embeddings to compute class prototypes in each run, while we use all image embeddings but vary the model initialization seed across runs.

Since the vision and language models generally produce embeddings of different dimensionalities, in this experiment, we employ dual-pass GCN encoders without weight sharing. While this design accommodates modality-specific feature spaces, it also makes the alignment task more challenging, as the model must learn to reconcile heterogeneous latent representations.

## C.6. Hyperparameters

We adopt a similar hyperparameter configuration for the vision-language benchmarks. Specifically, we set $k = 5$ when constructing the $k$-NN graphs. Each modality is encoded with a 4-layer dual-pass GCN encoder, and we use 9 Laplacian eigenvectors for CIFAR-10 and CINIC-10, and 90 eigenvectors for CIFAR-100 and ImageNet-100. The hidden and output dimensions of the encoder are both set to 512. All other hyperparameters follow the general settings described in Section C.3.

### C.6.1. BASELINES

We compare against a set of established solvers and heuristics for the alignment problem. LocalCKA (Maniparambil et al., 2024) leverages the centered kernel alignment (CKA) metric to approximate the QAP with a linear assignment formulation, providing an efficient method for vision–language correspondence. Optimal Transport (OT) methods (Peyré et al., 2016) address the alignment by modeling embeddings as probability distributions and computing the minimal transport cost, thereby preserving geometric structure across modalities. The Fast Approximate QAP algorithm (FAQ) (Vogelstein et al., 2015) is a well-known primal heuristic that relaxes the QAP and iteratively refines the solution, yielding scalable but approximate alignments. MPOpt (Hutschenreiter et al., 2021) represents a generic mathematical programming approach, solving the alignment as a constrained optimization problem using standard formulations. Gurobi (Gurobi Optimization, LLC, 2023) is a commercial off-the-shelf solver for mixed-integer and quadratic programs, providing near-optimal results for small problem instances. Finally, the Hahn-Grant solver (Schnaus et al., 2025) is a dual ascent algorithm that produces strong lower bounds by repeatedly solving linear assignment problems.

We also reference two recent vision–language alignment methods. Vec2Vec (Cai et al., 2021) maps embeddings from different models into a shared latent space using input/output adapters, a shared backbone, and adversarial plus structural losses. CycleReward (Bahng et al., 2025) learns vision–language alignment via cycle-consistency-based preference data and a reward model. These models are not designed for graph alignment and therefore are not direct competitors. Moreover, a full evaluation would require reproducing their results on our benchmarks, which is beyond the scope of this work and is deferred to future studies focused specifically on this domain.

### C.6.2. VISION AND LANGUAGE MODELS

We adopt the set of 32 vision models used in (Schnaus et al., 2025). For self-supervised methods, we use DINO (Caron et al., 2021) models (`RN50` and `ViT-S/B` with patch sizes 16 and 8) trained on ImageNet-1k and DINOv2 (Oquab et al., 2023) models (`ViT-S/B/L/G` with patch size 14) trained on the LVD-142M dataset, as well as fully supervised models such as DeiT variants (Touvron et al., 2021) (Tiny, Small, and Base with patch size 16, including distilled and high-resolution @384 versions) and ConvNeXt models (Liu et al., 2022b) (Base and Large, pretrained on ImageNet-1k or ImageNet-22k, with additional fine-tuned @384 variants). For vision–language pretraining, we employ CLIP (Ramesh et al., 2022) with both ResNet backbones (`RN50`, `RN101`, `RN50x4`, `RN50x16`, `RN50x64`) and Vision Transformer architectures (`ViT-B/32`,

`ViT-B/16`, `ViT-L/14`, `ViT-L/14@336`). All experiments are conducted using official implementations and pretrained weights, ensuring consistent and reliable representation extraction for each model.

We consider four pretrained language models spanning diverse architectures and training paradigms, including the `RN50x4` model from CLIP (Ramesh et al., 2022) and three models, `all-MiniLM-L6-v2`, `all-mpnet-base-v2`, and `all-Roberta-large-v1`, extracted from the SentenceTransformers library (Reimers & Gurevych, 2019). All vision and language models used in our experiments are summarized in Table 5.

*Table 5.* Summary of vision and language models used in the experiments

| **Vision models** |
| --- |
| **DINO** (Caron et al., 2021): RN50, ViT-S/16, ViT-S/8, ViT-B/16, ViT-B/8 |
| **DINOv2** (Oquab et al., 2023): ViT-S/14, ViT-B/14, ViT-L/14, ViT-G/14 |
| **DeiT** (Touvron et al., 2021): DeiT-T/16, DeiT-T/16d, DeiT-S/16, DeiT-S/16d, DeiT-B/16, DeiT-B/16@384, DeiT-B/16d, DeiT-B/16d@384 |
| **ConvNeXt** (Liu et al., 2022b): CN-B-1, CN-B-22, CN-L-1, CN-L-22, CN-L-22ft@384, CN-XL-22ft@384 |
| **CLIP** (Ramesh et al., 2022): RN50, RN101, RN50x4, RN50x16, RN50x64, ViT-B/32, ViT-B/16, ViT-L/14, ViT-L/14@336 |

| **Language models** |
| --- |
| **CLIP** (Ramesh et al., 2022): RN50x4 |
| **SentenceTransformers** (Reimers & Gurevych, 2019): all-MiniLM-L6-v2, all-mpnet-base-v2, all-Roberta-large-v1 |

## D. Ablation analysis

To analyze the impact of individual components in the proposed framework, we conduct an ablation study evaluating variants with specific modules removed or modified. We compare GADL against: (1) **GADL w/ GCN encoder**: replaces the dual-pass GCN with a standard single-pass GCN while retaining all other components; (2) **GADL w/ low-pass only**: replaces the dual-pass encoder with the low-pass branch alone ($\mathbf{Z} = \mathbf{Z}_l$) while retaining all other components; (3) **GADL w/ high-pass only**: replaces the dual-pass encoder with the high-pass branch alone ($\mathbf{Z} = \mathbf{Z}_h$) while retaining all other components; (4) **GADL w/o bijectivity regularization**: removes the bijectivity regularization term while keeping the dual-pass encoder; (5) **GADL w/o orthogonality regularization**: omits orthogonality regularization while maintaining all other components, and (6) **GADL w/o latent-space alignment**: removes both bijectivity and orthogonality regularizations, relying solely on the dual-pass encoder without any geometric constraints on the latent spaces.

Results are summarized in Table 6. They highlight the individual contribution of each component to the overall alignment performance. Replacing the dual-pass GCN with a standard encoder causes substantial accuracy drops.

*Table 6.* Performance of GADL and its variants on real-world graph alignment benchmarks.

| Method | ACM-DBLP | | | | Douban Online-Offline | | | |
| --- | --- | --- | --- | --- | --- | --- | --- | --- |
| | Hit@1 | Hit@5 | Hit@10 | Hit@50 | Hit@1 | Hit@5 | Hit@10 | Hit@50 |
| GADL w/ GCN encoder | 81.68 | 92.22 | 95.24 | 97.88 | 43.38 | 62.96 | 71.1 | 88.55 |
| GADL w/ low-pass only | 81.62 | 92.2 | 95.18 | 97.27 | 42.12 | 62.27 | 70.57 | 88.25 |
| GADL w/ high-pass only | 79.17 | 90.45 | 92.4 | 94.66 | 40.25 | 60.5 | 69.88 | 87.37 |
| GADL w/o bijectivity regularization | 88.47 | 94.6 | 96.06 | 98.37 | 52.68 | 72.89 | 80.14 | 94.78 |
| GADL w/o orthogonality regularization | 88.51 | 94.48 | 96.06 | 98.35 | 51.96 | 72.8 | 79.51 | 94.72 |
| GADL w/o latent-space alignment | 87.42 | 93.5 | 96.03 | 98.34 | 49.35 | 70.23 | 76.72 | 92.66 |
| GADL | 88.63 | 94.76 | 96.16 | 98.41 | 53.31 | 73.61 | 80.67 | 94.18 |

The results reveal a clear performance hierarchy across encoder variants that is consistent with our theoretical analysis. The high-pass only variant performs worst on both datasets, as discarding smooth structural context produces noisy embeddings that are difficult to geometrically align via the functional map module. The low-pass only variant performs nearly identically to the standard GCN, which is expected since both capture the same smooth neighborhood-aggregating behavior. The full dual-pass encoder, combining both branches, consistently achieves the best performance, validating that the two spectral components provide complementary information as established in Theorem 3.2.

The results show that the dual-pass encoder provides greater improvement on Douban than ACM-DBLP, reflecting the impact of initial node features on encoder effectiveness. Essentially, the Douban dataset contains sparse, high-dimensional node features with many zero entries, causing standard GCN embeddings to become overly smooth and less discriminative.

Consequently, the dual-pass filters lead to a significant improvement in matching accuracy. In contrast, the ACM-DBLP features are denser and more informative, so the standard GCN already generates sufficiently distinctive embeddings, with the high-pass component providing only moderate improvement.

Moreover, the results indicate that removing both regularizers leads to a clear performance degradation across benchmarks, with a more pronounced impact on Douban Online-Offline than on ACM-DBLP. This discrepancy can be attributed to several factors: 1) Douban Online-Offline consists of heterogeneous graph sources, one from online social interactions and one from offline event co-attendance, with inherently misaligned structures and dynamics, making latent space communication more critical. In contrast, ACM-DBLP contains two co-authorship networks from similar academic databases with more comparable structural properties. 2) The Offline graph is much smaller and sparser than the Online graph. This structural mismatch causes embeddings to naturally drift into different geometric spaces during training, making the bijectivity and orthogonality constraints more valuable for alignment. 3) As noted earlier, Douban has sparse, high-dimensional features with many zero entries. This sparsity, combined with structural differences, means that without explicit geometric constraints, the latent spaces can diverge significantly. The regularizations help anchor these spaces together despite the feature sparsity.

In essence, the larger improvements on heterogeneous graphs with structural and feature inconsistencies empirically validate that our latent-space alignment framework is most effective where it is most needed, directly confirming our motivation for designing this module to address the core challenges outlined in the introduction.

## E. Spectral energy distribution of dual-pass encoder

To empirically validate the analytical frequency responses derived in Section 3.2, Figure 5 reports the normalized spectral energy $\|\Phi_k^\top \mathbf{Z}\|^2$ per eigenvector of the symmetric normalized graph Laplacian $\Phi$, computed from the trained embeddings $\mathbf{Z}_l$ and $\mathbf{Z}_h$. The results show a clear visual contrast: the low-pass branch energy concentrates near $\tilde{\lambda} \approx 0$, confirming it captures smooth, globally consistent structure, while the high-pass branch energy concentrates near $\tilde{\lambda} \approx 2$, confirming it captures fine-grained local differences between neighboring nodes. Together, these results empirically confirm that the two branches learn complementary representations directly from data, consistent with the analytical frequency responses derived in Section 3.2.

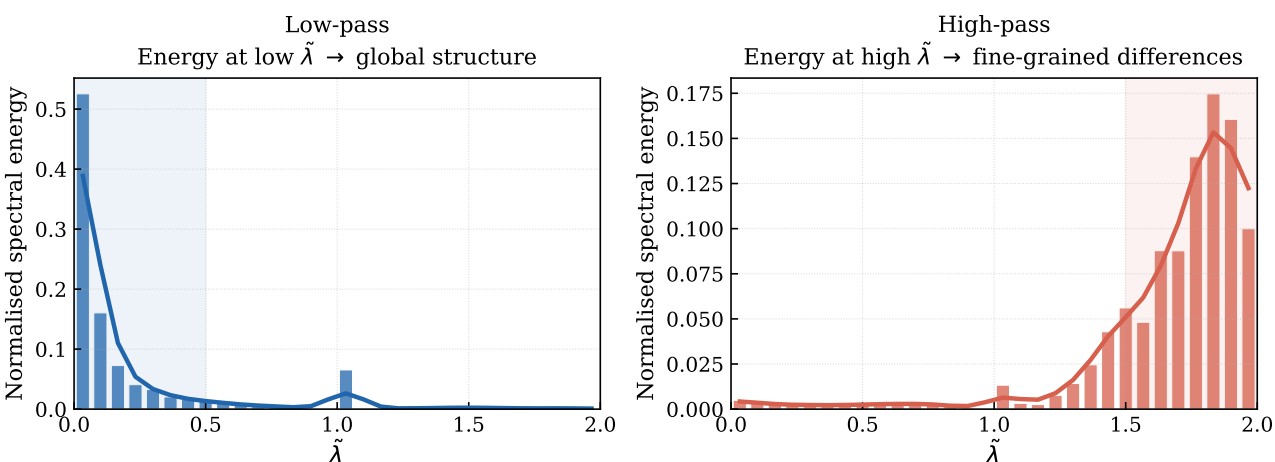

*Figure 5.* Spectral energy distribution of node embeddings (Douban Online-Offline)

## F. Ablation study on graph encoders

We evaluate the impact of different GNN encoder architectures on the alignment accuracy within the GADL framework using two real-world benchmark datasets. Specifically, we conduct a comparative evaluation of our proposed dual-pass GCN encoder against four GNN variants: GCN (Kipf & Welling, 2017), GIN (Xu et al., 2019), JKGNN (Xu et al., 2018), and TIDE (Behmanesh et al., 2023). For these encoders, we adopt a 6-layer architecture with ReLU activation functions, following the configurations presented in their respective papers. Additionally, TIDE is applied in a single-channel setup, where the learnable parameter $t$ is shared across all channels.

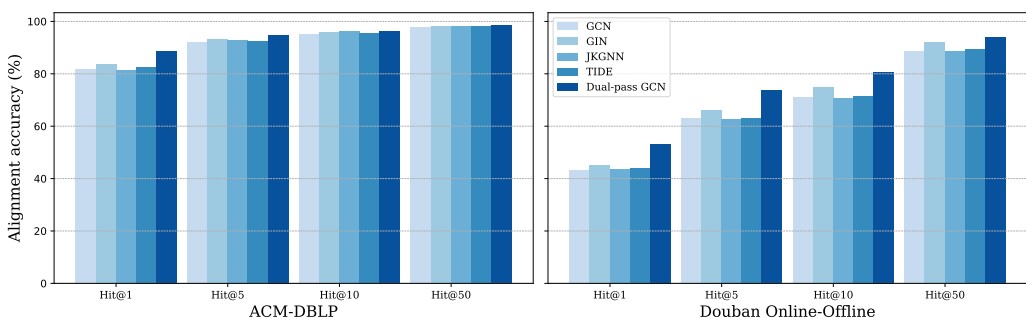

*Figure 6.* Encoder comparison on graph alignment performance (Hit@$K$).

As shown in Figure 6, the proposed dual-pass GCN achieves consistently higher alignment accuracy across both datasets. Among the others, the GIN encoder performs best because it is designed to better capture graph structure by extending the Weisfeiler-Lehman (WL) graph isomorphism test, which helps it distinguish nodes more effectively. Since more expressive node representations reduce ambiguity in identifying correct correspondences, this enhanced expressiveness directly contributes to improved node alignment accuracy.

## G. Node discriminability

Since the benchmarks are entirely unsupervised and lack node labels, we design a label-free experiment to verify that the dual-pass encoder preserves node discriminability: a node uniqueness ratio measuring the mean nearest-neighbour distance in embedding space, where higher values indicate greater node discriminability. Table 7 reports the node uniqueness ratio across five benchmarks, comparing GCN, low-pass only, high-pass only, and dual-pass encoders.

*Table 7.* Node uniqueness ratio across five benchmarks. Higher values indicate greater node discriminability.

| Dataset | Encoder | $Z_1$ (source) | $Z_2$ (target) | Mean | Rank |
|---|---|---|---|---|---|
| Douban Online-Offline | GCN | 0.0749 | 0.0630 | 0.0689 | 4 |
| | Low-pass only | 0.2976 | 0.1949 | 0.2463 | 3 |
| | High-pass only | 0.4233 | 0.3225 | 0.3729 | 1 |
| | Dual-pass | 0.3740 | 0.2599 | 0.3169 | 2 |
| ACM-DBLP | GCN | 0.1308 | 0.1402 | 0.1355 | 3 |
| | Low-pass only | 0.1299 | 0.1394 | 0.1346 | 4 |
| | High-pass only | 0.1387 | 0.1556 | 0.1471 | 1 |
| | Dual-pass | 0.1313 | 0.1407 | 0.1360 | 2 |
| Arenas (0.05) | GCN | 0.0029 | 0.0065 | 0.0047 | 4 |
| | Low-pass only | 0.0174 | 0.0169 | 0.0172 | 3 |
| | High-pass only | 0.0415 | 0.0395 | 0.0405 | 1 |
| | Dual-pass | 0.0255 | 0.0247 | 0.0251 | 2 |
| Cora (0.05) | GCN | 0.0659 | 0.0805 | 0.0732 | 4 |
| | Low-pass only | 0.0860 | 0.0866 | 0.0863 | 3 |
| | High-pass only | 0.1050 | 0.1063 | 0.1057 | 1 |
| | Dual-pass | 0.1028 | 0.1046 | 0.1037 | 2 |
| Douban (0.05) | GCN | 0.0296 | 0.0291 | 0.0293 | 3 |
| | Low-pass only | 0.0267 | 0.0263 | 0.0265 | 4 |
| | High-pass only | 0.0504 | 0.0581 | 0.0543 | 1 |
| | Dual-pass | 0.0349 | 0.0344 | 0.0347 | 2 |

The dual-pass encoder ranks 2nd in every benchmark, consistently outperforming both the standard GCN and low-pass only, confirming that the high-pass branch meaningfully enhances discriminability. Crucially, the dual-pass encoder never degrades below the low-pass baseline, empirically validating that the high-pass branch adds complementary discriminative information.

Importantly, this metric reflects only embedding spread, and a higher score does not translate directly into better alignment accuracy. High-pass filtering amplifies all high-frequency differences indiscriminately, including structural noise and

cross-graph inconsistencies, producing embeddings that are spread out but geometrically inconsistent across graphs. The dual-pass encoder balances discriminability with neighbourhood preservation, yielding embeddings that are both sufficiently unique for node identification and structurally consistent for cross-graph matching (Theorem 3.2), which directly explains the superior Hit@1 accuracy reported in Table 2.

## H. Oversmoothing analysis of the dual-pass encoder

We measure MAD (mean average distance, i.e., the average L2 distance between embeddings of neighboring nodes) across GNN depth on real-world benchmarks, comparing six configurations: standard GCN, GCN + PairNorm (Zhao & Akoglu, 2020), GCN + DropEdge (Rong et al., 2020), low-pass only, high-pass only, and dual-pass (all embeddings are L2-normalised before computing MAD). Results are shown in Figure 7.

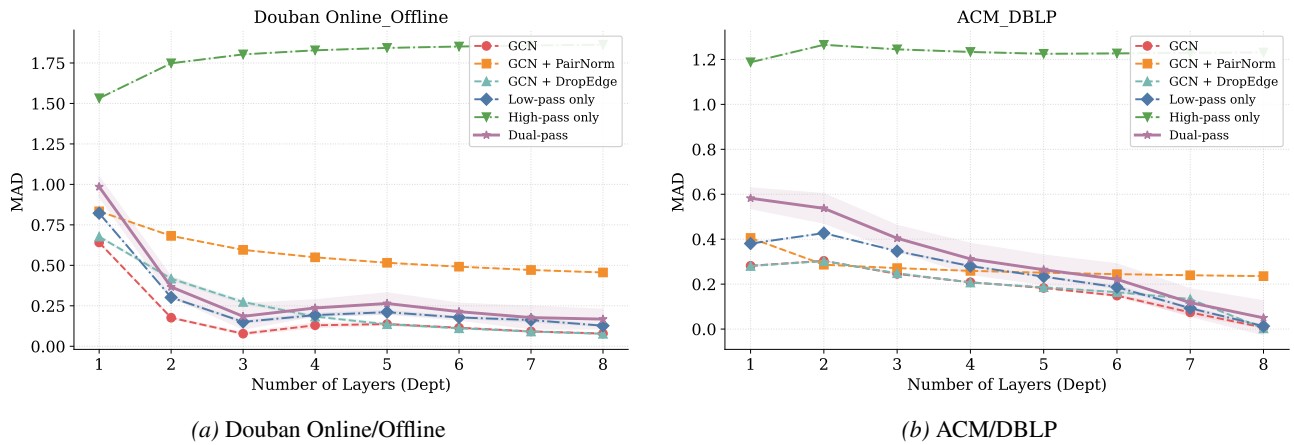

*(a)* Douban Online/Offline  *(b)* ACM/DBLP

*Figure 7.* MAD vs Depth Across GNN Variants on real-world benchmarks

The high-pass model achieves the highest MAD by a large margin, and GCN + PairNorm also maintains relatively high MAD on Douban Online-Offline. However, this does not translate into better graph alignment performance: MAD captures only intra-graph separation, whereas graph alignment requires cross-graph consistency. The high-pass model increases MAD by amplifying high-frequency components, which separates neighboring nodes but destroys the global structural information needed for alignment. Similarly, GCN + PairNorm increases MAD through per-graph normalisation, spreading node embeddings within each graph but centering them around different means, resulting in misaligned embedding spaces across graphs and making corresponding nodes harder to match. As a result, both methods achieve high MAD yet fail to produce embeddings suitable for accurate graph alignment.

Dual-pass achieves a moderate and stable MAD that neither collapses like standard GCN or low-pass only, nor artificially inflates like high-pass only or PairNorm. This balance reflects the core design principle of our encoder: the low-pass branch preserves global structure for cross-graph correspondence, while the high-pass branch maintains node-level discriminability and prevents embedding collapse.

## I. Computational Complexity Analysis

The computational complexity of GADL comprises four main components: the dual-pass spectral encoder, the functional map module, the bijectivity and orthogonality regularizers, and the node-alignment step.

The encoder employs two complementary GCN branches, each requiring sparse matrix propagation and feature transformation with per-layer complexity $\mathcal{O}(|\mathcal{E}|d + |\mathcal{V}|d^2)$, where $|\mathcal{V}|$ and $|\mathcal{E}|$ denote the number of nodes and edges, respectively, and $d$ is the embedding dimension. For $k$ layers, the dual-pass encoder incurs total cost $\mathcal{O}(2k|\mathcal{E}|d + 2k|\mathcal{V}|d^2) = \mathcal{O}(k|\mathcal{E}|d + k|\mathcal{V}|d^2)$, which simplifies to $\mathcal{O}(k|\mathcal{V}|d)$ for sparse graphs where $|\mathcal{E}| = \mathcal{O}(|\mathcal{V}|)$ and $d \ll |\mathcal{V}|$.

The functional map module computes transformations $\mathbf{C}_{12}, \mathbf{C}_{21} \in \mathbb{R}^{d \times d}$ between latent spaces and applies them to node embeddings, with a total complexity of $\mathcal{O}(|\mathcal{V}|d^2)$, arising from the matrix multiplications of the functional maps with the node embeddings. Notably, this cost excludes eigendecomposition, as our method relies on precomputed eigenvalues and

eigenvectors.

The bijectivity and orthogonality regularizers each require $\mathcal{O}(d^3)$ operations for matrix multiplication and Frobenius norm computation, contributing negligible overhead.

The alignment stage computes pairwise node similarities via inner products with complexity $\mathcal{O}(|\mathcal{V}|^2 d)$, followed by greedy Hungarian algorithm with complexity $\mathcal{O}(|\mathcal{V}|^2)$.

Combining all components, GADL has total complexity $\mathcal{O}(k|\mathcal{E}|d + k|\mathcal{V}|d^2 + |\mathcal{V}|d^2 + d^3 + |\mathcal{V}|^2)$.

For sparse graphs where $|\mathcal{E}| = \mathcal{O}(|\mathcal{V}|)$ and moderate embedding dimensions $d \ll |\mathcal{V}|$, this simplifies to $\mathcal{O}(|\mathcal{V}|^2)$, dominated by the node-alignment step.

### I.1. Runtime Evaluation

We evaluate the computational efficiency of GADL against state-of-the-art methods across six benchmark datasets of varying scales. Table 8 reports the training time (in seconds) per epoch.

*Table 8.* Training time comparison (seconds per epoch)

| Dataset | Graph Size | WAlign | T-GAE | SLOTAlign | GADL |
|---|---|---|---|---|---|
| Celegans | Small | 0.04 | 0.18 | 0.22 | 0.08 |
| Douban | Medium | 0.24 | 1.16 | 1.27 | 0.54 |
| Douban Online-Offline | Medium | 0.17 | 0.28 | 0.54 | 0.22 |
| ACM-DBLP | Large | 0.63 | 2.75 | 3.25 | 1.57 |
| Dblp | Large | 8.51 | 22.46 | – | 11.55 |
| Coauthor CS | Large | 9.52 | 29.28 | – | 16.73 |

Note: "–" indicates timeout after 1 hour training.

GADL vs. SLOTAlign. GADL significantly outperforms the optimization-based SLOTAlign, which requires iterative alternating optimization with complexity $\mathcal{O}(T \cdot (|\mathcal{V}_1|^2|\mathcal{V}_2| + |\mathcal{V}_1||\mathcal{V}_2|^2))$ (Tang et al., 2023a). While sparsity can reduce SLOTAlign to $O(|\mathcal{V}_1||\mathcal{V}_2|(d_1 + d_2) + |\mathcal{V}_1|l_2 + |\mathcal{V}_2|l_1)$, it still remains significantly more expensive than GADL. As a result, SLOTAlign fails to complete within reasonable time on larger graphs (Dblp, Coauthor CS).

GADL vs. T-GAE. While both methods exhibit $\mathcal{O}(|\mathcal{V}|^2)$ complexity dominated by node matching, GADL demonstrates 1.5-2× faster runtime across all datasets. This speedup stems from lightweight dual-pass GCN encoder compared to the deeper GIN architecture used in T-GAE (6-12 layers). The functional map module adds only minimal overhead, as its $\mathcal{O}(|\mathcal{V}|d^2)$ complexity (with $d \ll |\mathcal{V}|$) is negligible compared to the $\mathcal{O}(|\mathcal{V}|^2)$ cost of node matching.

GADL vs. WAlign. WAlign achieves efficient runtime by employing a lightweight GCN encoder and replacing the Sinkhorn-based optimal transport with simple pairwise similarity computation and greedy matching, resulting in $\mathcal{O}(|\mathcal{V}|^2)$ complexity. This makes it significantly faster than optimization-based methods like SLOTAlign. However, this efficiency reduces alignment accuracy and robustness, as its performance drops under perturbations (Table 1).

**Eigendecomposition overhead.** The eigendecomposition of the graph Laplacian is excluded from the per-epoch runtime comparison in Table 8 because it is computed once as a preprocessing step before training begins and is never recomputed during training or inference. Its cost is therefore a fixed one-time overhead that does not affect per-epoch training time, which is the standard convention in the functional map literature.

To directly address this overhead, we measured the actual eigendecomposition time on each benchmark, ranging from 1s on smaller graphs (Celegans) to 45s on larger graphs (Coauthor CS). Even when including this one-time cost in the total runtime, computed as preprocessing time + (number of epochs × per-epoch time), and ignoring preprocessing costs required by other methods (e.g., precomputed structural features in T-GAE), GADL remains faster than both T-GAE and SLOTAlign. This overhead is small relative to total training time and negligible on larger graphs where per-epoch cost dominates.

## J. Hyperparameter sensitivity

We conduct a sensitivity analysis to examine the influence of key hyperparameters on model performance. In particular, we focus on the loss weighting coefficients $\lambda_{\mathrm{FM}}$, $\lambda_{\mathrm{bij}}$, and $\lambda_{\mathrm{orth}}$, which control the relative importance of the functional map loss, bijectivity constraint, and orthogonality regularization, respectively, in the overall training objective.

To isolate the effect of each hyperparameter, we vary its value over a defined range while keeping the remaining parameters fixed at their optimal values, as determined through prior validation. We evaluate model performance using the Hit@1 accuracy metric on both real-world benchmark datasets.

The results in Figure 8 demonstrate how model performance responds to variations in each hyperparameter, using the optimal values identified through tuning as a reference. The results demonstrate that our approach is stable across a wide range of settings, while also pointing out where tuning is most important for best results. These insights provide practical guidance for selecting effective hyperparameter configurations when applying the model to new benchmarks.

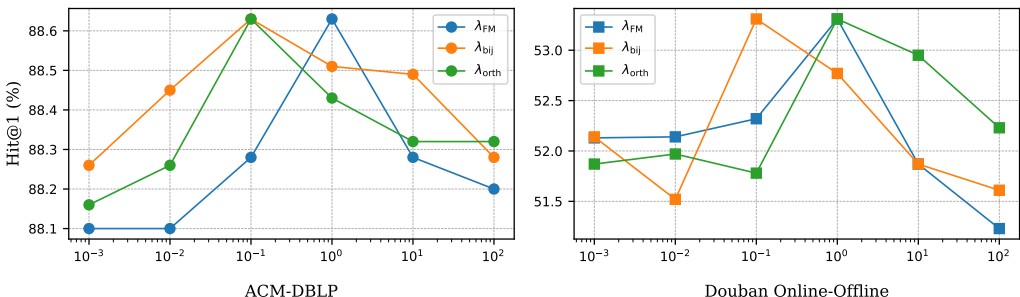

*Figure 8.* Hyperparameter sensitivity analysis on ACM-DBLP and Douban Online-Offline datasets. We report Hit@1 accuracy while varying each hyperparameter ($\lambda_{\mathrm{FM}}$, $\lambda_{\mathrm{bij}}$, $\lambda_{\mathrm{orth}}$) independently.

*Table 9.* Vision-language alignment accuracies on CIFAR-100 and ImageNet-100 with two language models (best-over-training).

| Model | CIFAR-100 | | ImageNet-100 | |
|---|---|---|---|---|
| | all-mpnet-base-v2 | All-Roberta-large-v1 | all-mpnet-base-v2 | All-Roberta-large-v1 |
| **CLIP** | | | | |
| RN50x16 | 46.67 ± 2.87 | 67.00 ± 4.97 | 40.67 ± 0.94 | 63.67 ± 3.40 |
| RN50x64 | 48.33 ± 0.47 | 76.33 ± 0.94 | 37.33 ± 0.47 | 58.00 ± 4.55 |
| ViT-L/14 | 46.00 ± 7.87 | 85.67 ± 1.89 | 74.00 ± 0.82 | 61.00 ± 1.41 |
| ViT-L/14@336 | 79.67 ± 2.05 | 81.00 ± 1.41 | 41.33 ± 8.26 | 45.33 ± 14.20 |
| **DeiT** | | | | |
| DeiT-B/16 | 47.00 ± 0.82 | 84.00 ± 0.82 | 67.33 ± 0.47 | 58.67 ± 4.03 |
| DeiT-B/16@384 | 54.67 ± 5.44 | 88.00 ± 0.00 | 35.33 ± 0.47 | 57.67 ± 7.54 |
| DeiT-B/16d | 58.33 ± 8.63 | 58.67 ± 15.22 | 38.33 ± 6.02 | 54.33 ± 7.04 |
| DeiT-B/16d@384 | 47.33 ± 8.99 | 42.67 ± 6.34 | 67.33 ± 4.50 | 65.67 ± 0.94 |
| **DINOv2** | | | | |
| ViT-B/14 | 48.33 ± 0.47 | 55.00 ± 6.53 | 83.67 ± 0.47 | 60.33 ± 0.94 |
| ViT-S/14 | 48.67 ± 4.64 | 58.33 ± 3.22 | 35.67 ± 0.47 | 63.67 ± 4.50 |
| ViT-L/14 | 79.67 ± 1.70 | 60.67 ± 5.19 | 48.33 ± 2.49 | 69.67 ± 7.13 |
| ViT-G/14 | 67.67 ± 1.70 | 58.33 ± 0.47 | 44.33 ± 8.26 | 49.33 ± 0.47 |

## K. Comprehensive results for vision-language model combinations

To provide a comprehensive evaluation of vision–language alignment, we test our proposed model across multiple vision and language model combinations on two benchmark datasets, CIFAR-10 and CINIC-10. The vision models considered include CLIP (Ramesh et al., 2022), ConvNeXt (Liu et al., 2022b), DINO (Caron et al., 2021), DINOv2 (Oquab et al., 2023), and DeiT (Touvron et al., 2021). For the language models, we include the `RN50x4` model from CLIP (Ramesh et al., 2022) as well as three models from the SentenceTransformers library (Reimers & Gurevych, 2019): `all-MiniLM-L6-v2`, `all-mpnet-base-v2`, and `all-Roberta-large-v1`.

The results are summarized in Figure 9, where the error bars indicate the standard deviation computed over 20 random seeds.

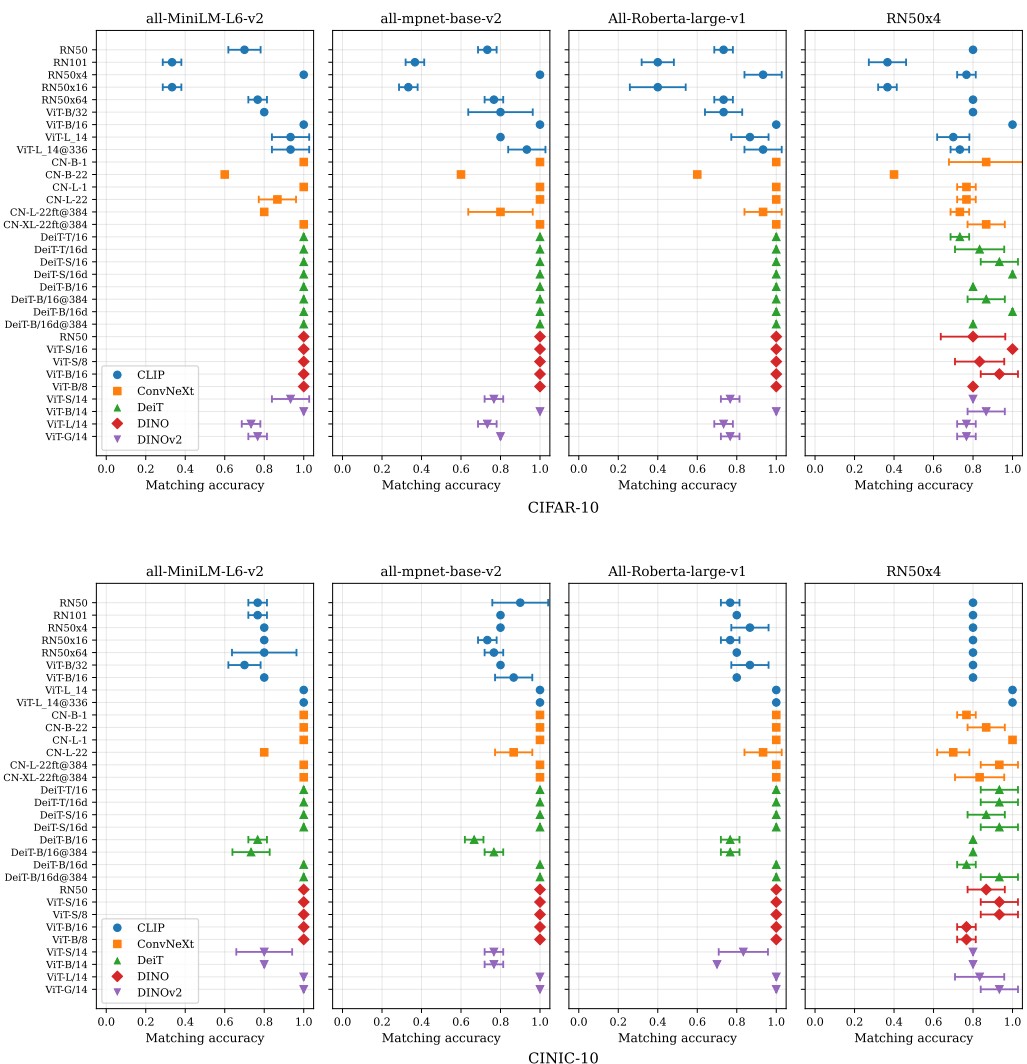

*Figure 9.* Vision–language accuracy (best-over-training) of the proposed model on combinations of multiple vision models with four language models on CIFAR-10 (top row) and CINIC-10 (bottom row)

These results indicate that our proposed approach consistently achieves higher matching accuracies across diverse vision–language encoder combinations, outperforming state-of-the-art baselines such as the Hahn-Grant solver in most configurations (cf. Figure 4 in the Hahn-Grant paper (Schnaus et al., 2025)).

Our method shows particularly strong performance with DINO and DINOv2 models, where most configurations achieve matching accuracies above 0.8 on both CIFAR-10 and CINIC-10 datasets. The CLIP models also demonstrate competitive performance, with several variants reaching near-perfect accuracy. Several models achieve perfect accuracy on CIFAR-10, including CLIP: `RN50x4` and `ViT-B/16`, all ConvNeXt variants except `CN-B-22`, most DeiT models, and all DINO models. Comparable trends are observed on CINIC-10, where numerous models also reach 100% accuracy. The results indicate that the choice of pre-training model has a greater influence on performance than model size. DINO models exhibit remarkable consistency, achieving near-perfect accuracy in most configurations. In contrast, some larger models, such as CLIP: `RN101` and `RN50x16`, perform poorly (33–40% on CIFAR-10), indicating that model scale alone does not guarantee superior performance.

Among the different language encoders, the sentence-transformer models (`all-MiniLM-L6-v2`, `all-mpnet-base-v2`, and `All-Roberta-large-v1`) outperform `RN50x4`, as they are specifically opti-

mized for semantic text representation and generating high-quality text embeddings. In contrast, the `RN50x4` encoder in CLIP is trained with an objective that prioritizes vision–language alignment rather than producing rich text embeddings.

We further evaluate the proposed model across diverse vision–language combinations on the larger-scale CIFAR-100 and ImageNet-100 benchmarks. Table 9 summarizes the results. The reported numbers correspond to the best test accuracy achieved over training, providing an optimistic upper bound on performance. These results demonstrate that model performance is highly context-dependent, with no single architecture achieving universal superiority across CIFAR-100 and ImageNet-100.

