# OpenReview forum: "Graph Alignment via Dual-Pass Spectral Encoding and Latent Space Communication"
_ICML.cc/2026/Conference — ICML 2026 regular_

### Official Review · Reviewer_PWuk · 2026-03-06

**Soundness:** 3
**Presentation:** 3
**Significance:** 2
**Originality:** 2
**Overall Recommendation:** 4
**Confidence:** 3

**Summary:**

This paper investigates unsupervised graph alignment and proposes Graph Alignment via Dual-pass encoder and Latent-space communication (GADL) method. Previous graph alignment methods face two inherent challenges: (1) The similarity request for the embedding of neighboring nodes may lead to the degradation of node distinctiveness in GNN embeddings. (2) Each GAE independently produces a latent space for its respective graph, resulting in misaligned embeddings.

To tackle these challenges, GADL has two core designs: (1) a dual-pass spectral encoder that concatenates low-pass and high-pass GCN branches to balance structural smoothness and node distinctiveness; and (2) a geometry-aware functional map module that aligns the two latent spaces.

Experiments on semi-synthetic benchmarks with pertubs of different levels show higher or competitive alignment accuracy and robustness of GADL. On real-world datasets, GADL significantly outperforms baseline methods, demonstrating its practical value.

**Compliance With Llm Reviewing Policy:**

Affirmed.

**Key Questions For Authors:**

See weaknesses.

**Limitations:**

yes

**Strengths And Weaknesses:**

Strengths

1. GADL is well-motivated and specific; it analyzes two inherent flaws in the graph alignment task and proposes a dual-pass spectral encoder and bijective/isometric functional maps, respectively, to address them.
2. Experimental results demonstrate that GADL achieves performance superior to baseline methods, and can generaize to vision-language alignment tasks beyond graph domains

Weaknesses

1. Combining low-pass and high-pass filters has been explored in prior work; Does applying this idea to graph alignment task to address the over-smoothing problem have some unique research challenge?
2. Experimental results show that although GADL surpasses other baseline methods in accuracy, it does not demonstrate sufficient advantages in robustness compared to T-GAE and SLOTAlign, thereby challenging the necessity of latent space communication.

---

> ### Author Rebuttal · Authors · 2026-03-31
>
> We sincerely thank the reviewer for their feedback. Below, we address each concern.
>
> >w1. Combining low-pass and high-pass filters ...
>
> As mentioned in the last paragraph of Appendix A, we have carefully reviewed these prior works (e.g., GPR-GNN, BernNet, TFE-GNN, FAGCN) and agree that spectral filtering on graphs has been well explored. However, we would like to clarify that our approach differs in several fundamental ways:
>
> 1. We filter in **both feature space and map space** (a fundamental distinction). Graph alignment requires resolving a dual challenge: node features must be sharp (high-frequency) for distinctiveness, while the correspondence itself must be smooth for robustness and structural consistency. Prior works that combine low-pass and high-pass filtering attempt to balance both objectives purely in the feature space. Our key insight is that these objectives should instead be enforced through complementary filtering in both feature and map spaces: we employ a dual-pass encoder to inject high-frequency discriminability into the features, paired with a functional map module that acts as a low-pass filter on the correspondence itself. This design, simultaneous filtering in both feature and map spaces, is entirely absent from prior literature, and is a key contribution of our method.
>
> 2. We acknowledge that our dual-pass encoder can be viewed as a principled adaptation of spectral filtering specifically tailored to the **requirements of graph alignment**: the explicit split into analytically defined complementary frequency bands directly promotes the two properties established in Theorem 3.2 (neighborhood preservation and enhanced node discriminability). Crucially, this encoder is designed to work in concert with the functional map module, and it is this combination, filtering in both feature and map spaces simultaneously, that constitutes our core contribution.
>
> 3. Prior spectral GNN methods are primarily designed for single-graph tasks such as node classification, and their filter coefficients are optimized for within-graph representation quality. **They do not address the cross-graph challenges** central to alignment: ensuring geometric consistency between independently learned latent spaces and enforcing smoothness on the correspondence itself. Our architecturally fixed filters are a deliberate design choice motivated by these alignment-specific requirements rather than by the supervision regime alone.
>
> In summary, our contribution lies not in the filters themselves, but in the principled insight that graph alignment requires simultaneously filtering in both feature and map spaces, a perspective and design that prior spectral GNN works do not address.
>
>
> >w.2. Experimental results...
>
> Although the improvements over T-GAE and SLOTAlign are modest in only a few settings in Table 1, GADL consistently outperforms both methods across the vast majority of benchmarks and perturbation levels. Moreover, this comparison alone does not capture the true contribution of latent space communication, two points directly support its necessity:
>
> 1. The ablation study in Table 6 (Appendix D) directly validates the necessity of latent space communication: removing both regularizers (GADL w/o latent-space alignment) leads to a clear performance drop across all benchmarks, with a particularly pronounced impact on Douban Online-Offline, a heterogeneous dataset where latent spaces are most likely to diverge without explicit geometric constraints. This directly confirms that latent space communication is not redundant but most effective precisely where it is most needed: in settings with structural and feature inconsistencies across graphs.
>
> 2. GADL consistently outperforms T-GAE and SLOTAlign across nearly every dataset and perturbation level in Table 1, including the most challenging 5% perturbation scenarios. Latent space communication provides reliable geometric grounding that becomes increasingly valuable under challenging conditions, preventing the performance collapse observed in methods without explicit alignment constraints (a property directly confirmed by the ablation).
>
> For more clarification, we include a t-SNE visualization of the learned node embeddings and compare embeddings obtained without and with the latent space alignment (functional map module). Please see Figure 2 in the anonymized attachment: [anonymized attachment (Fig. 2)](https://anonymous.4open.science/r/Anonymous-5407/anonymous_rebuttal_figures.pdf).
>
> This figure shows that latent space alignment (functional map module) reduces cross-graph misalignment. Each colored circle and triangle represents a ground-truth node pair (50 pairs shown), connected by a line. Left: lines are long and scattered, showing high misalignment. Right: lines are shorter, pairs are closer, and nodes cluster more coherently, demonstrating improved geometric consistency across graphs.

---

### Official Review · Reviewer_4KRn · 2026-03-07

**Soundness:** 2
**Presentation:** 2
**Significance:** 3
**Originality:** 3
**Overall Recommendation:** 4
**Confidence:** 4

**Summary:**

This paper proposes GADL (Graph Alignment with Dual-pass Encoder and Latent Space Communication), which aims to address the problem of unsupervised graph alignment. The method combines a dual-pass GCN encoder with latent space functional maps, forming a unified framework that simultaneously tackles the two key challenges of node discriminability degradation and latent space misalignment.

**Compliance With Llm Reviewing Policy:**

Affirmed.

**Final Justification:**

The authors have addressed most of my concerns. I am willing to increase my score.

**Key Questions For Authors:**

Please refer to the Weaknesses.

**Limitations:**

Please refer to the Weaknesses.

**Strengths And Weaknesses:**

Strengths:

1.	This paper integrates graph encoding, latent space functional mapping, and regularization constraints to propose a complete and well-structured framework for unsupervised graph alignment.

2.	The experimental results are reliable and robust, with comprehensive validation on both semi-synthetic and real-world datasets. The inclusion of vision–language cross-modal alignment further demonstrates the generalization capability of the method.

Weaknesses:

1.	The authors claim that GADL simultaneously addresses node discriminability degradation and latent space misalignment, but the experiments only report the final alignment accuracy, lacking direct evidence for the intermediate mechanisms. Could latent space alignment be visualized via node embeddings? Alternatively, can node classification or intra-/inter-class difference metrics be used to verify that node discriminability is preserved?

2.	The authors claim that the low-pass branch captures global structure while the high-pass branch preserves fine-grained differences, yet there is no theoretical proof or spectral analysis. Could frequency response curves or spectral energy distribution experiments be provided to support this claim?

3.	The experimental section lacks ablation studies specifically verifying the contributions of the low-pass and high-pass branches, making it unclear how effective each branch is.

4.	The high-pass branch is proposed to prevent node embedding over-smoothing, but no experiments verify this. Could the authors show embedding variance or node similarity curves across layers, compared with existing over-smoothing mitigation methods such as PairNorm or DropEdge?

5.	In the runtime and matching time comparisons, the eigendecomposition in the functional map module is excluded. Why is it not included in the total training time? If eigendecomposition is counted, the overall complexity of GADL may be higher. Would it still have an advantage compared to other methods?

---

> ### Author Rebuttal · Authors · 2026-03-31
>
> We sincerely thank reviewer for their thoughtful feedback.
>
> >W1.1.Could latent space alignment be visualized ...?
>
> We include a t-SNE visualization of the learned node embeddings and compare embeddings obtained without and with the functional map module. Please see Fig. 2 in the anonymized attachment: [anonymized attachment (figures+tabels)](https://anonymous.4open.science/r/Anonymous-5407/anonymous_rebuttal_figures.pdf)
>
> This figure shows that latent space alignment (functional map module) reduces cross-graph misalignment. Each colored circle and triangle represents a ground-truth node pair (50 pairs shown), connected by a line. Left: lines are long and scattered, showing high misalignment. Right: lines are shorter, pairs are closer, and nodes cluster more coherently, demonstrating improved geometric consistency across graphs.
>
> >W1.2 can node classification or intra-/inter-class difference metrics be used to ...?
>
> Since our benchmarks are unsupervised, we evaluate the dual-pass encoder using a label-free node uniqueness ratio (mean nearest-neighbor distance; higher = more discriminative; please see Table 1 in the attachment).
>
> Dual-pass ranks 2nd in every benchmark, always outperforming both Simple GCN and Low-pass only, confirming that the high-pass branch meaningfully enhances discriminability. Crucially, dual-pass never degrades below the low-pass baseline, empirically validating that the high-pass branch adds complementary discriminative information.
>
> We would like to point out that this metric only reflects embedding spread, and the higher score does not translate into better alignment accuracy.
>
> >W.2
>
> We will add a figure showing the frequency response curves of all three filters over λ∈[0,2]  (Fig. 1 in the attachment).
>
> The low-pass branch is strictly decreasing, smoothly attenuating high-frequency components while fully preserving structure-aware signals near λ≈0. The high-pass branch is strictly increasing, suppressing low frequencies and emphasizing fine-grained discriminative variations near λ≈2. In contrast, the standard GCN filter becomes negative for λ~>1(shaded region), potentially causing instability during training.
>
> Additionally, we add a spectral energy experiment to support this claim (Fig. 3 in the attachment). We project the trained low-pass and high-pass embeddings onto the Laplacian eigenvectors and measure normalized energy per eigenvector. Low-pass energy concentrates near λ≈0, capturing smooth, global structure, while high-pass energy concentrates near λ≈2, capturing fine-grained local differences. This demonstrates that the two branches learn complementary global and local representations from real data, consistent with the analytical frequency responses in Section 3.2.
>
> >W.3
>
> The ablation study in Appendix D (Table 6) already evaluates architectural contributions. We updated the table by changing the first row to GADL with GCN encoder (replacing dual-pass with single-pass GCN) and adding two variants: GADL with low-pass only and high-pass only, isolating each spectral branch’s effect (Table 2 in attachment).
>
> Results show a consistent performance hierarchy. High-pass only performs worst, losing smooth structural context. Low-pass only performs nearly identically to the standard GCN, which is expected since both capture the same smooth neighborhood-aggregating behavior. Dual-pass consistently performs best, confirming the two spectral branches provide complementary information (Theorem 3.2).
>
> >W.4.
>
> We would like to clarify that the dual-pass encoder aims to produce embeddings that are both structure-aware and discriminative, not solely to reduce oversmoothing. We measured MAD (average L2 distance between neighboring node embeddings) across GNN depth for six configurations: GCN, GCN + PairNorm, GCN + DropEdge, Low-pass only, High-pass only, and Dual-pass (Fig. 4 in the attachment).
>
> Results show the High-pass model achieves the highest MAD, and GCN + PairNorm maintains relatively high MAD on Douban Online-Offline. However, higher MAD does not improve graph alignment: High-pass separates neighbors but loses global structure, while PairNorm spreads embeddings per graph but misaligns spaces. Both yield high MAD yet fail at accurate cross-graph matching
>
> Dual-pass achieves a moderate, avoiding collapse like GCN/Low-pass only and overinflation like High-pass/PairNorm. This balance reflects our design: the low-pass branch preserves global structure for alignment, while the high-pass branch maintains node discriminability.
>
> >W.5.
>
> The graph Laplacian eigendecomposition is excluded from per-epoch runtimes because it is computed once as preprocessing. Its cost is small (1 s on Celegans, 45 s on Coauthor CS) and negligible compared to total training, defined as preprocessing time + (number of epochs × per-epoch time). Even including it, GADL remains faster than T-GAE and SLOTAlign (1718 s vs. 2928 s on Coauthor CS). Updated runtimes will be added to the manuscript.

---

> > ### Author Rebuttal · Reviewer_4KRn · 2026-04-01
> >
> > I appreciate that the authors provide substantial additional experiments and analysis, which address most of my concerns. However, some aspects, such as the formal demonstration of the causal effect of the low-pass and high-pass branches on alignment performance and their role in mitigating over-smoothing, remain not fully convincing.

---

> > > ### Author Response · Authors · 2026-04-02
> > >
> > > We thank the reviewer for the follow-up. We believe the combination of theoretical, spectral, and empirical evidence provided in our rebuttal collectively and rigorously addresses this concern. We decompose our answer into four complementary components:
> > >
> > > (Please find figures and tables here: [Rebuttal Figures](https://anonymous.4open.science/r/Anonymous-5407/anonymous_rebuttal_figures.pdf))
> > >
> > > 1. **Causal effect of each branch on alignment performance.** The ablation study (Table 2) evaluates the contribution of each encoder variant to alignment accuracy. High-pass only achieves the lowest alignment accuracy; Low-pass only matches standard GCN in accuracy. Their combination (dual-pass) achieves the best alignment accuracy, confirming that discriminability (high-pass) and structural consistency (low-pass) are both necessary for alignment, removing either degrades a different aspect and neither alone is sufficient.
> > >
> > > 2. **Role of each branch in mitigating over-smoothing.** MAD (Mean Average Distance between neighbouring node embeddings, higher values = less smoothing) experiments (Fig. 4) show that dual-pass maintains more stable MAD across depth compared to GCN and low-pass only. This is a direct consequence of the filter design (Fig. 1): $p_h​$ is strictly monotonically increasing, guaranteeing high-frequency amplification at every layer, preventing the uniform embedding convergence that $p_l$​ alone cannot avoid. We respectfully emphasise that over-smoothing mitigation itself is not a claim of this paper- if it were sufficient for alignment, high-pass only would perform well, yet it achieves the highest MAD and the lowest alignment accuracy simultaneously.
> > >
> > > 3. **Formal guarantee of spectral complementarity on alignment improvement.** Theorem 3.2 formally guarantees $\Delta_\text{dual} \geq \max(\Delta_\text{low}, \Delta_\text{high}) + \gamma$, where $\Delta$ is the separation margin between corresponding and non-corresponding node pairs, meaning dual-pass achieves strictly better node correspondence discrimination than either branch alone.
> > >
> > > 4. **Empirical evidence of spectral complementarity.** Frequency response curves (Fig. 1) show $p_l$​ and $p_h$ are strictly monotonically decreasing and increasing, partitioning the spectral domain into complementary regions. Spectral energy experiments (Fig. 3) confirm this holds on real trained embeddings: low-pass energy concentrates near $λ≈0$, high-pass energy near $λ≈2$. Together, these empirically confirm that the two branches learn complementary representations, **directly supporting** the alignment improvement in point 1 and the over-smoothing behaviour in point 2.
> > >
> > > We hope the reviewer finds this evidence sufficient and that our responses across all questions have addressed the raised concerns. We are grateful for the reviewer's suggestions, the additional experiments and analysis have meaningfully strengthened both the clarity and the scientific rigor of our contributions, and we hope this will be reflected in an updated score.

---

### Official Review · Reviewer_fh5Y · 2026-03-09

**Soundness:** 2
**Presentation:** 3
**Significance:** 3
**Originality:** 3
**Overall Recommendation:** 5
**Confidence:** 4

**Summary:**

The paper proposes GADL, a framework for unsupervised graph alignment that identifies corresponding nodes across two graphs without known correspondences. It addresses two major challenges in embedding-based alignment: loss of node distinctiveness due to GNN oversmoothing and misaligned latent spaces across graphs. The method uses a dual-pass GCN encoder combining low-pass and high-pass spectral filters to produce node embeddings that preserve both structural context and discriminative features. To align embeddings across graphs, the model introduces a geometry-aware functional map module that learns transformations between latent spaces while enforcing bijectivity and orthogonality constraints. Experiments on multiple graph benchmarks and vision–language alignment tasks show that the proposed method improves robustness and alignment accuracy compared to prior approaches.

**Compliance With Llm Reviewing Policy:**

Affirmed.

**Final Justification:**

After reading the other reviews and the authors’ rebuttal, I believe the paper has clear merit. The proposed method introduces a spectral approach to graph alignment, and demonstrates empirical improvements over prior work. The rebuttal addresses my main concerns raised during the review process and helps clarify the scope and contributions of the method.

Overall, I find the work to be technically sound and relevant to the community, especially for spectral graph approaches. I therefore support acceptance.

**Key Questions For Authors:**

### 1. Biijection formulation
The framework introduces a bijectivity regularization term that encourages the functional maps $C_{12}$ and $C_{21}$ to be inverses of each other, suggesting a one-to-one correspondence between nodes. However, the paper also assumes that the graphs may have different numbers of nodes $(|V_s| \neq |V_t|)$. Could the authors clarify how the bijectivity constraint is reconciled with this setting? Does the model effectively learn a partial matching, and if so, how is this reflected in the functional map formulation?

### 2. Functional map module's role in final alignment
It is not entirely clear how the functional map module contributes to the final node assignment. The paper introduces functional maps $C_{12}$ and $C_{21}$ to align latent spaces during training, but the final correspondences are obtained by computing cosine similarity between the embeddings and performing greedy matching, without explicitly applying the learned functional maps during inference. Could the authors clarify how the learned functional maps influence the final alignment?

**Limitations:**

The authors acknowledge some limitations of the proposed method, particularly the additional computational overhead introduced by the dual-pass encoder and the need for careful hyperparameter tuning. While these aspects are discussed in the paper, further analysis of the method’s behavior in settings with larger graphs or more heterogeneous structures could help better understand its practical applicability.

**Strengths And Weaknesses:**

## Soundness:
The paper presents a technically sound approach. The experimental results are comprehensive, demonstrating the method's robustness and improved alignment accuracy. However, some clarifications are needed regarding the reconciliation of the bijectivity constraint with graphs of different sizes and the role of the functional maps in the final alignment process.

## Presentation:
The paper is clearly written and generally easy to follow. The introduction provides a good motivation for the problem and the proposed approach, and the method is well positioned with respect to prior work. The technical sections are detailed and the experimental setup is well documented.

## Significance:
The paper addresses an important problem in graph alignment, which has applications in various domains. By combining spectral filtering with latent space alignment, the proposed method provides a potentially useful perspective on how to improve the robustness of embedding-based alignment methods. The approach could influence future work on graph representation learning and cross-graph correspondence.

## Originality:
The paper proposes a novel combination of spectral filtering and functional map–based alignment for graph alignment. In particular, the use of a dual-pass GCN encoder to combine low-pass and high-pass spectral components, together with a geometry-aware functional map module, is an interesting design that aims to address both node distinguishability and latent space misalignment. This combination differentiates the proposed method from most existing embedding-based alignment approaches.

---

> ### Author Rebuttal · Authors · 2026-03-30
>
> We sincerely thank reviewer for their thoughtful feedback.
>
> > 1. Bijictivity formulation
>
> We clarify that the bijectivity constraint operates in the **spectral domain**, not the node domain, and is therefore fully compatible with graphs of different sizes.Specifically, the functional maps C12, C21 are square matrices that operate on a shared spectral basis of fixed dimension r, regardless of the number of nodes in each graph. Node embeddings are projected onto the first r eigenvectors of their respective graph Laplacians, yielding spectral descriptors, which are projections of node embeddings onto the first r eigenvectors of each graph Laplacian. The functional map then aligns these r-dimensional spectral representations. Since both spectral bases have the same dimensionality r by construction, the bijectivity constraint is always well-defined and mathematically consistent, even when ∣Vs∣≠∣Vt∣∣.
>
> Importantly, bijectivity here enforces that the **linear operator between function spaces is invertible**, not that nodes are matched one-to-one. This prevents degenerate map solutions and encourages geometric consistency between latent spaces, serving as a structural prior in map space.
>
> We further note that bijectivity in the spectral domain is a well-established strategy in the functional maps literature, fully compatible with meshes of different connectivity and even strong partiality. Specifically, SURFMNet [1] employs spectral bijectivity for shape matching across meshes with different vertex counts, and DPFM [2] shows it remains effective even when large portions of one shape have no counterpart in the other. These works confirm that spectral bijectivity constrains the operator between function spaces, not the node-level correspondence, and is therefore well-suited to our setting where graphs may differ in size.
>
> Node-level matching is handled separately via cosine similarity and greedy assignment, which naturally accommodates partial and asymmetric correspondences when ∣Vs∣≠∣Vt∣. This design follows the established functional maps framework for 3D shape correspondence, where spectral bijectivity and partial pointwise matching coexist by construction.
>
>
> [1] Roufosse et al., CVPR 2019, Unsupervised Deep Learning for Structured Shape Matching
>
> [2] Attaiki et al., 3DV 2021, DPFM: Deep Partial Functional Maps
>
> > 2. Functional map modules in final alignment
>
> We clarify that the functional map module contributes to the final alignment **indirectly, through its influence on the learned embeddings** during training, rather than being applied explicitly at inference.
>
> During training, the functional map losses​, together with the bijectivity and orthogonality regularizations, act as geometric constraints that co-evolve with the encoder parameters. Specifically, since C12 and C21​ are trained end-to-end alongside the dual-pass encoder, the encoder is implicitly guided to produce embeddings whose spectral representations are mutually alignable by a bijective and isometric transformation. In fact, the functional map module shapes the geometry of the latent spaces during training so that corresponding nodes are pushed closer together and non-corresponding nodes are pushed apart.
>
> At inference, the encoder produces embeddings that already reflect this geometrically consistent structure, so cosine similarity between the resulting embeddings is sufficient to recover reliable correspondences without needing to explicitly apply C12​ or C21​.
>
> The functional map module can therefore be understood as a **geometry-aware training objective** that regularizes the embedding space, with its effect fully absorbed into the encoder weights by the end of training.
>
> Additionally, we would like to refer to [1], which shows that, under two mild conditions, feature completeness and basis-aligned functional maps, extracting a point-to-point map from a functional map is equivalent to nearest-neighbor search in feature space
>
> [1] Attaiki et al., Understanding and Improving Features Learned in Deep Functional Maps, CVPR 2023
>
> We will add a clarifying remark in the paper to make this distinction explicit.
>
>
> > Limitation
>
> Our real-world benchmarks already include **heterogeneous** settings, most notably Douban Online-Offline, which pairs graphs from fundamentally different social interaction modalities (online vs. offline), and large-scale graphs, such as ACM-DBLP, which involves large co-authorship networks with structural and feature discrepancies.
>
> Regarding scalability, we note that a key challenge in graph alignment is the lack of large-scale benchmarks with reliable ground truth. Most existing datasets are small to medium in size or provide only partial node correspondences, limiting evaluation at truly large scale.
>
> We will clarify this, the lack of more diverse benchmarks, in the limitation section of the paper, and highlight that constructing or adopting larger and more diverse benchmarks is an important direction for future work.

---

> > ### Author Rebuttal · Reviewer_fh5Y · 2026-04-01
> >
> > I thank the authors for the clarifications, and I confirm my score

---

> > > ### Author Response · Authors · 2026-04-02
> > >
> > > We sincerely thank the reviewer for their careful reading of our work and for taking the time to engage with our responses. **We are glad that our clarifications were satisfactory** and appreciate the constructive feedback, which has helped us improve the clarity and presentation of the paper.

---

### Official Review · Reviewer_UwoU · 2026-03-18

**Soundness:** 2
**Presentation:** 2
**Significance:** 3
**Originality:** 2
**Overall Recommendation:** 3
**Confidence:** 4

**Summary:**

Self supervised methods for graph learning require to match similar nodes in different graphs. The authors identify 2 possible obstacles to this: oversmoothing and latent space misalignment. They propose a "dual pass" net using laplacian features to correct this (exploiting the noise robustness of Laplacian eigenfunctions). The low and high pass filters satisfy their requirements of an ideal node embedding for alignment as shown in theorem 3.2. Also, a map identifying the 2 latent spaces is learned to address the second challenge.

**Compliance With Llm Reviewing Policy:**

Affirmed.

**Key Questions For Authors:**

How did you come to identify the 2 challenges of node matching? A short paragraph on this would help understanding. Possibly easy examples too.

**Limitations:**

yes

**Strengths And Weaknesses:**

Strength:
The authors identification of the 2 challenges of node matching are a success. They are intuitive and the fixes they propose make sense. There may be more challenges to discover.

Weaknesses:
The abstracts and intro dont do a good job of describing what they do and this made the paper very difficult to read. Giving a better early description of the proposed architecture or at least the concepts it was designed around would improve the presentation. I have to get to page 4 to learn what a dual pass network is. Thats hard on the reader. You dont really have to explain it before in depth but explaining why you need low and high pass is critical. Theorem 3.2 should be mentioned informally before also.
I would like to see  a quick description of the experiments at the begining of section 4. I am not familiar with these tasks. 4.1 mentions adding noise but not what is measured.

---

> ### Author Rebuttal · Authors · 2026-03-30
>
> We thank the reviewer for their constructive feedback. We agree that the concerns are primarily related to presentation and clarity, and we will ensure that all comments are fully incorporated in the final version. We believe these changes are straightforward and can be addressed with minor revisions, without altering the core technical contributions.
>
> Concretely, we will make the following revisions:
> - Revise the abstract and introduction to include an intuitive description of the dual-pass architecture and its motivation.
> - Introduce an early explanation of the need for low- and high-pass filtering.
> - Provide an informal preview of Theorem 3.2 in the introduction.
> - Add a summary of experimental settings at the beginning of Section 4.
> - Clarify evaluation metrics and noise modeling in Section 4.1.
>
> > Question: How did you come to identify the 2 challenges of node matching? A short paragraph on this would help understanding. Possibly easy examples too.
>
> The two challenges are motivated by fundamental properties of GNN-based embedding methods, supported by both theoretical analysis and empirical observation.
>
> **Challenge 1 (node distinctiveness)** follows directly from the spectral interpretation of GNN aggregation.
>
> For clarify, in the final version, we will add a plot showing the frequency responses of all filters over $\tilde{\lambda} \in [0,2]$. The frequency response curves show how each filter responds to different eigenvalues (Please see Figure 1 in the anonymized attachment: [anonymous_rebuttal_figures](https://anonymous.4open.science/r/Anonymous-5407/anonymous_rebuttal_figures.pdf))
>
> - **The low-pass** branch $p_l(\tilde{\lambda}) = 1 - \frac{1}{2}\tilde{\lambda}$ is strictly monotonically decreasing, smoothly attenuating high-frequency components while fully preserving the smooth, structure-aware signals at $\tilde{\lambda} \approx 0$.
>
> - **The high-pass** branch $p_h(\tilde{\lambda}) = \frac{1}{2}\tilde{\lambda}$ is strictly monotonically increasing, suppressing low-frequency signals while amplifying fine-grained discriminative variations at $\tilde{\lambda} \approx 2$.
>
> - In contrast, the **standard GCN filter** $p_{GCN}(\tilde{\lambda}) = 1 - \tilde{\lambda}$ becomes negative for $\tilde{\lambda} > 1$ (shaded region), introducing instability. Additionally, GCN acts approximately as a low-pass filter, repeated neighborhood aggregation inevitably smooths node representations, reducing their discriminability. This is a well-known limitation in the GNN literature (oversmoothing), but its specific impact on alignment, where unique node identification is critical, has been underexplored.
>
> **Challenge 2 (latent space misalignment)** emerges from the unsupervised nature of the problem. Even with shared encoder weights, independently optimized embeddings for two different graphs are not guaranteed to occupy geometrically compatible spaces. This is empirically demonstrated in Figure 1, where we show that: (i) feature noise, (ii) structural perturbations, and (iii) different random initializations each cause corresponding nodes to diverge significantly in latent space, even on identical graphs.
>
> We will add a clarifying paragraph in the introduction making the motivation for both challenges more explicit.

---

### Decision · Program_Chairs · 2026-04-30

**Decision:**

Accept (regular)

**Comment:**

**Overall Assessment**

The paper received mostly positive but borderline reviews, with three reviewers recommending acceptance and one recommending rejection. Overall, the paper presents a well-motivated and technically sound framework for unsupervised graph alignment. While some aspects would benefit from clearer justification and stronger differentiation from prior work, the contributions are solid and the empirical performance supports acceptance.

**Summary of Contributions**

The paper proposes GADL, a framework for unsupervised graph alignment that aims to identify corresponding nodes across two graphs without access to known node correspondences. The work addresses two key challenges common to embedding-based graph alignment methods: (1) loss of node distinctiveness due to GNN oversmoothing, and (2), misalignment of latent spaces across different graphs.
To mitigate oversmoothing, the authors introduce a dual-pass GCN encoder that combines low-pass and high-pass spectral filters, enabling node embeddings to capture both global structural information and discriminative local features. To address latent space misalignment, the method incorporates a geometry-aware functional map module that learns mappings between the latent spaces of two graphs while enforcing bijectivity and orthogonality constraints. Experimental results across multiple graph alignment benchmarks, as well as vision language alignment tasks, demonstrate improved alignment accuracy and robustness compared to prior methods.

**Strengths**

> The paper is clearly written, well organized, and generally easy to follow.

> Graph alignment is a fundamental problem with wide-ranging applications, and improving unsupervised alignment remains a meaningful challenge.

> The framework is thoughtfully designed, with each component directly motivated by identified shortcomings in existing methods.

> The experimental evaluation is extensive and shows consistent improvements in alignment accuracy across benchmarks.

**Weaknesses**

> Further discussion is needed on how bijectivity is reconciled when aligning graphs of different sizes, as well as on the precise role of the functional maps in producing the final node correspondences. Authors are asked to implement the reviewer’s comments in this respect.

> While combining low-pass and high-pass filters has been explored in prior work, it is not entirely clear whether adapting this idea to graph alignment introduces unique challenges or insights beyond existing techniques.

> Although GADL improves alignment accuracy, its robustness advantages over strong baselines such as T-GAE and SLOTAlign are less pronounced.

**Final Evaluation**

Despite the above weaknesses, this paper makes a strong and coherent contribution to the graph alignment literature. The proposed framework is well motivated, technically sound, and empirically validated, and it meaningfully advances the state of unsupervised graph alignment.